# COVID-19 Vaccines over Three Years after the Outbreak of the COVID-19 Epidemic

**DOI:** 10.3390/v15091786

**Published:** 2023-08-23

**Authors:** Aleksandra Anna Zasada, Aniela Darlińska, Aldona Wiatrzyk, Katarzyna Woźnica, Kamila Formińska, Urszula Czajka, Małgorzata Główka, Klaudia Lis, Paulina Górska

**Affiliations:** Department of Sera and Vaccines Evaluation, National Institute of Public Health NIH—National Research Institute, 00-791 Warsaw, Poland; adarlinska@pzh.gov.pl (A.D.); awiatrzyk@pzh.gov.pl (A.W.); kwoznica@pzh.gov.pl (K.W.); kforminska@pzh.gov.pl (K.F.); uczajka@pzh.gov.pl (U.C.); mglowka@pzh.gov.pl (M.G.); klis@pzh.gov.pl (K.L.); pgorska@pzh.gov.pl (P.G.)

**Keywords:** COVID-19, vaccine, mRNA vaccine, vector-based vaccine, DNA vaccine, VLP vaccine, vaccine quality, vaccine safety

## Abstract

The outbreak of COVID-19 started in December 2019 and spread rapidly all over the world. It became clear that the development of an effective vaccine was the only way to stop the pandemic. It was the first time in the history of infectious diseases that the process of the development of a new vaccine was conducted on such a large scale and accelerated so rapidly. At the end of 2020, the first COVID-19 vaccines were approved for marketing. At the end of March 2023, over three years after the outbreak of the COVID-19 pandemic, 199 vaccines were in pre-clinical development and 183 in clinical development. The candidate vaccines in the clinical phase are based on the following platforms: protein subunit, DNA, RNA, non-replication viral vector, replicating viral vector, inactivated virus, virus-like particles, live attenuated virus, replicating viral vector combined with an antigen-presenting cell, non-replication viral vector combined with an antigen-presenting cell, and bacterial antigen-spore expression vector. Some of the new vaccine platforms have been approved for the first time for human application. This review presents COVID-19 vaccines currently available in the world, procedures for assurance of the quality and safety of the vaccines, the vaccinated population, as well as future perspectives for the new vaccine platforms in drug and therapy development for infectious and non-infectious diseases.

## 1. Introduction

The outbreak of COVID-19 started in late December 2019 in Wuhan, China, as ‘atypical pneumonia’ of unknown etiology in a cluster of patients [1]. A new human coronavirus was identified, initially named 2019-nCoV and later renamed as severe acute respiratory syndrome coronavirus 2 (SARS-CoV-2) [2].

The outbreak spread rapidly and at the end of February 2020 over 90 countries, areas, or territories were affected. On March 11, 2020, the World Health Organization declared the COVID-19 pandemic.

According to WHO data, 8,148,864 cases of COVID-19 and 481,517 deaths were registered at the end of June 2020. The number of cases and deaths would grow rapidly, and at the end of 2020, there were 58,309,299 cases and 1,375,045 deaths registered worldwide. A total of 185 countries, areas, or territories were affected by COVID-19 at the end of 2020. It became clear that the development of an effective vaccine was the only way to limit the transmission of the virus and limit the long-term complications and deaths caused by the SARS-CoV-2 infection and stop the pandemic. It was the first time in the history of infectious diseases that the process of the development of a new vaccine was conducted on such a large scale and accelerated so rapidly. At the end of the first quarter of 2020, two candidate COVID-19 vaccines were in phase I clinical trials and 60 in the pre-clinical phase. Nearly a month later, five candidate COVID-19 vaccines were in the clinical trials phase, including one in phase II and seventy-one in the pre-clinical phase. At the end of 2020, the first vaccines were approved for marketing. Figure 1 shows the COVID-19 vaccine development dynamics according to the draft landscape published by the WHO.

The acceleration of the COVID-19 vaccine development was possible, among others, thanks to prior knowledge gained during the SARS-CoV and MERS-CoV epidemics when the vaccine development started but was ceased due to the end of the epidemics [3]. Moreover, the mRNA platform was already being used for clinical trials and the adenovirus vector platform was utilized for the marketed vaccine against Ebola (Zabdeno/Mvabea). What is more, medicine authorization agencies, such as the Food and Drug Administration (FDA) and the European Medicines Agency (EMA), had already developed procedures for fast-track approval in emergency situations [4].

Standard vaccine research and development is a long process that usually takes 10–15 years. In this mode of action, the whole process is conducted in multiple sequential steps. In the emergency mode of vaccine development, the phases are simultaneously overlapped to speed up the whole process [4,5,6].

The expedited procedures of a medicine approval, called the Emergency Use Authorization (FDA) or Conditional Marketing Authorization (EMA), may be applied for a medicine that addresses an extraordinary medical need such as in the case of pandemics. The procedures require less comprehensive clinical data than normal but it has to be demonstrated that the benefit of the immediate availability of the medicine outweighs the risk inherent in the fact that there is still a demand for additional data. Once a conditional marketing authorization has been granted, the marketing authorization holder must fulfill specific obligations, according to the negotiated schedule, which include completing the ongoing or new studies or collecting additional data to confirm that the benefit–risk balance of the medicine remains positive [4], (https://www.ema.europa.eu/en/human-regulatory/marketing-authorisation/conditional-marketing-authorisation, accessed date 28 July 2023).

At the end of March 2023, over three years after the outbreak of the COVID-19 pandemic, 199 vaccines are in pre-clinical development and 183 in clinical development, including vaccines in phase IV clinical trials, i.e., post-marketing surveillance. The candidate vaccines in clinical phase are based on the following platforms: protein subunit, DNA, RNA, non-replication viral vector, replicating viral vector, inactivated virus, virus-like particles, live attenuated virus, replicating viral vector combined with an antigen-presenting cell, non-replication viral vector combined with an antigen-presenting cell, and bacterial antigen-spore expression vector. The majority of the candidate vaccines in the clinical phase are protein subunit vaccines (33%) and RNA vaccines (23%) (Figure 2). Vaccines approved for emergency use by at least one country have been presented in Table 1.

## 2. Current COVID-19 Vaccine Platforms

### 2.1. Live Attenuated and Inactivated Vaccines

Live attenuated vaccines have been successfully produced for decades to prevent the most dangerous diseases such as tuberculosis, measles, mumps, rubella, chickenpox, or rotavirus infection. A live attenuated vaccine usually produces immunity against all proteins of the target pathogen and therefore might be more effective against various genetic variants of the pathogen. However, live vaccines carry a certain risk of causing active disease in particularly vulnerable people, i.e., those with compromised immune systems. Currently, no live attenuated COVID-19 vaccine has been approved. However, two such vaccines are in the clinical phase of development: COVI-VAC produced by Codagenix and Serum Institute of India, which is in phase III clinical trials, and MV-014-2012 produced by Meissa Vaccines Inc., which is in phase I clinical trials. Both vaccines are administered intranasally (https://www.who.int/publications/m/item/draft-landscape-of-covid-19-candidate-vaccines, accessed date 10 March 2023).

The inactivated vaccines are prepared in a manner very similar to the live preparations. This type of vaccine contains pathogenic microorganisms, i.e., a virus or bacteria subjected to the inactivation process. During the preparation process, bacteria are multiplied on a suitable synthetic medium, while viruses are in a properly selected and prepared cell line. Inactivation involves killing microorganisms with appropriate chemicals such as phenol or formaldehyde, as well as with temperature or ionizing radiation. For each type of microorganism, there are strictly defined inactivation conditions (e.g., pH, temperature) and time needed to obtain a fully inactivated sample. After inactivation, the suspension should be sufficiently thickened and purified to eliminate any risk to the patient [7,8]. However, the production of inactivated vaccines requires special laboratory facilities to grow the virus or bacterium in safe conditions and can have a relatively long production time. Moreover, vaccines that are inactivated by the killing process reduce the potency of the antigen compared to live attenuated vaccines. Therefore, those vaccines require the use of an adjuvant in order to be effective. The most popular adjuvants, among those used so far, are aluminum compounds, such as aluminum hydroxide. The adjuvant on which the antigen is adsorbed allows reducing the rate of release of this antigen, which makes it possible to increase the post-vaccination response [9].

Currently, there have been 11 inactivated COVID-19 vaccines approved worldwide (https://www.who.int/publications/m/item/draft-landscape-of-covid-19-candidate-vaccines, accessed date 10 March 2023). The COVID-19 vaccine Covilo, vaccine developed by the Chinese company Sinopharm (Beijing) (also referred to as BBIBP-CorV (Vero cells), BIBP) is the most broadly approved inactivated COVID-19 vaccine. It has been approved in 93 countries: Algeria, Angola, Antigua and Barbuda, Argentina, Armenia, Bahrain, Bangladesh, Barbados, Belarus, Belize, (the Plurinational State of) Bolivia, Bosnia and Herzegovina, Brazil, Brunei Darussalam, Burkina Faso, Burundi, Cambodia, Cameroon, Chad, China, Comoros, Cuba, Côte d’Ivoire, Dominica, Dominican Republic, Egypt, Equatorial Guinea, Ethiopia, Gabon, Gambia, Georgia, Guinea, Guinea-Bissau, Guyana, Hungary, Indonesia, (the Islamic Republic of) Iran, Iraq, Jordan, Kazakhstan, Kenya, Kiribati, Kyrgyzstan, Lao People’s Democratic Republic, Lebanon, Madagascar, Malawi, Malaysia, the Maldives, Mauritania, Mauritius, Mexico, Mongolia, Montenegro, Morocco, Mozambique, Myanmar, Namibia, Nepal, Nicaragua, Niger, Nigeria, North Macedonia, Pakistan, Papua New Guinea, Paraguay, Peru, the Philippines, Republic of Moldova, Republic of the Congo, Rwanda, Senegal, Serbia, Seychelles, Sierra Leone, Solomon Islands, Somalia, South Africa, Sri Lanka, Sudan, Suriname, Thailand, Togo, Trinidad and Tobago, Tunisia, the United Arab Emirates, the United Republic of Tanzania, Vanuatu, (the Bolivarian Republic of) Venezuela, Vietnam, West Bank, Zambia, and Zimbabwe. The vaccine has been evaluated in 39 clinical trials in 18 countries (https://www.who.int/publications/m/item/draft-landscape-of-covid-19-candidate-vaccines, accessed date 10 March 2023). The vaccine contains two SARS-CoV-2 strains cultured in the Vero cell line and β-propanolide-inactivated WIV04 and HB02 and aluminum hydroxide as an adjuvant. All adverse reactions after the injection were of mild or moderate severity. The most common side effects observed were pain/tenderness at the injection site and fever. The vaccine efficacy was estimated as 72.8% for WIV04 and 78.1% for HB02 against the symptomatic coronavirus disease 2019 and 79% against severe disease or hospitalization [10,11,12]. The Covilo vaccine has been evaluated in 39 clinical trials in 18 countries. The vaccine is also manufactured by G42 Healthcare under the trade name Hayat-Vax (https://www.who.int/publications/m/item/draft-landscape-of-covid-19-candidate-vaccines, accessed date 10 March 2023). Moreover, the company Sinopharm (Wuhan) also developed an inactivated (Vero cells) vaccine that has been approved in two countries: China and the Philippines (https://www.who.int/publications/m/item/draft-landscape-of-covid-19-candidate-vaccines, accessed date 10 March 2023). The vaccine contains a SARS-CoV-2 strain WIV04 isolated from a patient in Wuhan. The virus was cultured in a Vero cell line and then inactivated with β-propiolactone twice. The vaccine is adsorbed on 0.5 mg aluminum adjuvant. The product contains sterile phosphate-buffered saline. It does not contain a preservative [10,11,12] (reactogenicity was seen within the first week after Sinopharm, Sputnik V, AZD1222, and COVIran Barekat vaccines: findings from the Iranian active vaccine surveillance system (published: 10 March 2023)).

Another Chinese company, Sinovac Life Sciences Co., developed an inactivated vaccine under the trade name CoronaVac. The product contains the CN02 strain of the SARS-CoV-2 virus chemically inactivated by β-propiolactone. The SARS-CoV-2 virus strain was cultured in the Vero cell line. Sinovac’s vaccine also contains an aluminum hydroxide adjuvant to enhance the immune response [13]. CoronaVac has been approved in 56 countries: Albania, Algeria, Argentina, Armenia, Azerbaijan, Bangladesh, Benin, Bolivia, Bosnia and Herzegovina, Botswana, Brazil, Burkina Faso, Cambodia, Chile, China, Colombia, Djibouti, Dominican Republic, Ecuador, Egypt, El Salvador, Georgia, Guinea, Hong Kong, Indonesia, Kazakhstan, Lao People’s Democratic Republic, Libya, Malawi, Malaysia, Mali, Mexico, Montenegro, Nepal, Oman, Pakistan, Panama, Paraguay, the Philippines, Republic of Moldova, Singapore, South Africa, Sri Lanka, Tajikistan, Thailand, Timor-Leste, Togo, Trinidad and Tobago, Tunisia, Turkey, Uganda, Ukraine, the United Republic of Tanzania, Uruguay, Venezuela, and Zimbabwe. The vaccine has been evaluated in 42 clinical trials in 10 countries (https://www.who.int/publications/m/item/draft-landscape-of-covid-19-candidate-vaccines, accessed date 10 March 2023). Most of the adverse reactions after injection were of mild or moderate severity. However, eight serious adverse reactions were identified, such as myopathy and colitis ulcerative. Efficacy differed from more than 50% in Brazil in protection against symptomatic and asymptomatic SARS-CoV-2 infection to 83.5% in Turkey, where the effectiveness in protection against symptoms of COVID-19 was assessed as well [14] (https://ec.europa.eu/health/documents/community-register/2020/20201221150522/anx_150522_en.pdf, accessed date 28 July 2023).

The VLA2001 is a vaccine developed by the Austrian company Valneva. It has been approved in 33 countries: Austria, Bahrain, Belgium, Bulgaria, Croatia, Cyprus, Czechia, Denmark, Estonia, Finland, France, Germany, Greece, Hungary, Iceland, Ireland, Italy, Latvia, Liechtenstein, Lithuania, Luxembourg, Malta, the Netherlands, Norway, Poland, Portugal, Romania, Slovakia, Slovenia, Spain, Sweden, the United Arab Emirates, and the United Kingdom of Great Britain and Northern Ireland. (https://www.who.int/publications/m/item/draft-landscape-of-covid-19-candidate-vaccines, accessed date 10 March 2023). The vaccine contains the Wuhan strain hCoV-19/Italy/INMI1-isl/2020, which was cultivated on Vero cells (African green monkey cells). The vaccine is adsorbed on two adjuvants: aluminum hydroxide and CpG1018 (cytosine phospho-guanine). It consists of a β-propiolactone inactivated whole virus with a high spike protein density. The combination of adjuvants helped to boost the immune response. All adverse reactions after injection were of mild or moderate severity. The most common side effects observed were pain/tenderness at the injection site, headaches, muscle pain, nausea, and vomiting [SCP]. The vaccine under the working name VLA2001 was well tolerated during the phase I and II clinical trials. Thus, its safety for patients has been demonstrated. People participating in the clinical trials, after administration of the product, developed antibodies against the SARS-CoV-2 spike protein as well as an immune response against other viral proteins. After two weeks, the levels of tested antibodies were as high as in people after infection with SARS-CoV-2 (EMA). All samples showed neutralizing antibodies against the original virus. The incidence of neutralizing antibodies against Omicron was 87% (https://valneva.com/press-release/valneva-reports-positive-phase-3-results-for-inactivated-adjuvanted-covid-19-vaccine-candidate-vla2001/, accessed date 30 March 2023). In 2021, the “Cov-Compare” trial, a phase III clinical trial, was launched to compare the safety and efficacy of Valneva’s vaccine with the previously approved AstraZeneca vaccine. The trials were conducted in patients over 18 years of age. It was proven to be more effective than the vector vaccine. The effectiveness of the preparation seemed to be sufficient for people aged 18 to 50 to protect against COVID-19.

The Covaxin vaccine developed by the Indian multinational biotechnology company Bharat Biotech (also referred to as BBV152) has been approved in 14 countries: Bahrain, Botswana, Guyana, India, Iran, Malaysia, Mauritius, Mexico, Nepal, Paraguay, the Philippines, Trinidad and Tobago, Vietnam, and Zimbabwe. The vaccine has been evaluated in 16 clinical trials in two countries (https://www.who.int/publications/m/item/draft-landscape-of-covid-19-candidate-vaccines, accessed date 10 March 2023). The vaccine contains whole virion, inactivated coronavirus (SARS-CoV-2) antigen (Strain: NIV-2020-770), and aluminum hydroxide gel as an adjuvant. Hypersensitivity reactions after the injection with BBV152 were rare and usually unserious. Severe allergic (anaphylactic) reactions have not been reported. The vaccine’s efficacy was estimated at 77.8% against symptomatic COVID-19 disease. Also, the efficacy against severe symptomatic COVID-19 disease is shown to be at the level of 93.4%. It has proven to neutralize the following variants: B.1.1.7 (Alpha), P.1-B.1.1.28 (Gamma) and P.2-B.1.1.28 (Zeta), B.1.617 (Kappa), B.1.351 and B.1.617.2 (Beta and Delta) (https://www.bharatbiotech.com/covaxin.html, accessed date 30 March 2023).

The KoviVac vaccine has been developed by the Chumakov Centre which is an Institute of the Russian Academy of Sciences. The product is an inactivated COVID-19 vaccine of limited use. It has been approved in three countries: Belarus, Cambodia, and the Russian Federation. The vaccine has been evaluated in three clinical trials in one country (the Russian Federation) (https://www.who.int/publications/m/item/draft-landscape-of-covid-19-candidate-vaccines, accessed date 10 March 2023). vaccine contains 3 μg or more of SARS-CoV-2 strain AYDAR-1 antigen inactivated by β-propiolactone and contains aluminum hydroxide as an adjuvant. The safety of the vaccine was evaluated in clinical trials, which proved good tolerability and safety. The most common adverse reaction was pain at the injection site. Any deaths and serious adverse events were not observed during those trials. The inactivated vaccine KoviVac exhibited a seroconversion rate of 86.9% in participants who were seronegative at the time of screening tests (https://www.vidal.ru/drugs/covivac, accessed date 30 March 2023).

The Research Institute for Biological Safety Problems in Kazakhstan developed QazCovid-in, commercially known as the QazVac vaccine. The vaccine has been approved in two countries: Kazakhstan and Kyrgyzstan (https://www.who.int/publications/m/item/draft-landscape-of-covid-19-candidate-vaccines, accessed date 10 March 2023). It is a formaldehyde-inactivated vaccine, adjuvanted with aluminum hydroxide, and administered intramuscularly in a two-dose scheme [15]. Adverse events within 7 days after the first vaccination included mild local and systemic reactions, which diminished significantly after the second dose. The preventive efficacy of the vaccination reached 82.0%. It was defined as the ratio of RT-PCR-confirmed COVID-19 cases relative to all vaccinated patients [16].

The KCONVAC vaccine developed by Shenzhen Kangtai Biological Products Co. (also referred to as SARS-CoV-2 Vaccine-Vero Cells) is an inactivated COVID-19 vaccine of limited use. It has been approved in two countries: China and Indonesia. The vaccine has been evaluated in seven clinical trials in one country (China) (https://www.who.int/publications/m/item/draft-landscape-of-covid-19-candidate-vaccines, accessed date 10 March 2023). The vaccine contains inactivated SARS-CoV-2 prepared on Vero cells. All adverse reactions after vaccination were of mild or moderate severity. The most common adverse reactions included headachess and pain at the injection site. The vaccine’s efficacy was estimated with the evaluation of the level of induced neutralizing antibodies and antigen-specific CD8 T cells after the booster vaccination. All results suggest that the vaccine is effective against the following variants: B.1.617.2 (Delta), (Alpha), B.1.351 (Beta), B.1.526.2, B.1.618, B.1.1.529 (Omicron), B.1.1.7, and B.1.617.3 [17,18].

Three inactivated COVID-19 vaccines have been approved only in the countries of its development: COVIran Barekat and FAKHRAVAC vaccines in Iran and the Turkovac vaccine in Turkey.

The COVIran Barekat vaccine was developed by the Iranian company Shifa Pharmed Industrial Co. (https://www.who.int/publications/m/item/draft-landscape-of-covid-19-candidate-vaccines, accessed date 10 March 2023). The vaccine contains the SARS-CoV-2 virus inactivated by β-propiolactone cultured on Vero monolayer cells. An aluminum adjuvant was used in the formulation of COVIran Barekat to achieve an effective and robust immune response [19]. All adverse reactions after the injection were of mild or moderate severity. The most common side effects observed included pain at the injection site and tiredness. Reactogenicity events were higher in women and younger people (reactogenicity was seen within the first week after Sinopharm, Sputnik V, AZD1222, and COVIran Barekat vaccines: findings from the Iranian active vaccine surveillance system (published: 10 March 2023)). The efficacy of the COVIran Barekat vaccine in preventing intensive care unit admission 3 months after the second dose reached 70%. The maximum effectiveness of the COVIran Barekat vaccine in preventing death 4 months after the administration of the second dose was over 90% [20].

The FAKHRAVAC vaccine was developed by the Organization of Defensive Innovation and Research in Iran (also referred to as MIVAC). The vaccine has been evaluated in three clinical trials in Iran (https://www.who.int/publications/m/item/draft-landscape-of-covid-19-candidate-vaccines, accessed date 10 March 2023). The vaccine contains isolates from oropharynx swabs collected from patients admitted to Iranian hospitals with confirmed COVID-19. The most common adverse reactions were headachess and pain at the injection site. The vaccine’s efficacy, such as increase in the neutralizing antibody titer two and four weeks after administration of the second dose, was estimated as 69.6% and 73.4% of the participants in the vaccinated group, respectively [21,22].

The Turkovac vaccine has been developed by the Health Institutes of Turkey (also referred to as ERUCOV-VAC) (https://www.who.int/publications/m/item/draft-landscape-of-covid-19-candidate-vaccines, accessed date 10 March 2023). The vaccine contains the inactivated SARS-CoV-2 antigen and aluminum hydroxide as an adjuvant. The safety of the vaccine was evaluated in eight clinical trials in Turkey. The vaccine exhibited good tolerability and safety. The most common adverse reactions were headachess, fatigue, and pain at the injection site. No deaths or serious adverse events were observed during the trials. The vaccine’s efficacy was estimated as the seroconversion rates for anti-SARS-CoV-2 spike antibodies at 94.3%. The neutralization assay yielded positivity rates of 51.2% [23].

### 2.2. Protein Subunit Vaccines

Subunit vaccines are the next-generation medicinal products following live and inactivated vaccines. They contain highly purified, immunogenic antigens, which are fragments of pathogenic microorganisms. The development of subunit vaccines has been based on the findings that administration of the entire pathogen is not a prerequisite for immune system stimulation. The purified antigenic fragment of the viral particle may act as a suitable inducer. Protein subunit vaccines are developed with the use of the protein components of viral particles. In addition, the combination of various antigens can be used as a conjugate vaccine [24,25,26,27]. Due to the fact that the antigenic composition is well-defined and tested, and there is no risk of incomplete virus inactivation or reversion to virulence, the risk of side effects is minimized, and such vaccines are considered very safe. However, subunit vaccines have low immunogenicity and require adjuvants to enhance the immunogenicity and, very often, booster doses [24,26,28]. Unlike a live attenuated vaccine, the immune response to a subunit vaccine is mostly antibody production with little or no cellular immunity results, whereas the immune response to a live attenuated vaccine is virtually identical to that produced by a natural infection (CDC, The Pink Book). Used adjuvants with various characteristics elicit distinctive immunological profiles with regard to the direction, duration, and strength of immune responses. Moreover, the use of adjuvants might stimulate not only humoral but also cellular immune responses [29]. Subunit vaccines are produced by recombined protein techniques using heterologous expression systems, prokaryotic or eukaryotic, such as bacterial, yeast, insect, or/and mammalian cells [13,30,31]. The antigen gene is introduced into yeast cells and grown in large fermentation bioreactors to produce recombinant protein subunits. After the purification process, the subunits are combined with preservatives and adjuvants [24,27]. In the COVID-19 vaccine development, the SARS-CoV-2 spike (S) protein is regarded as the key antigen. The protein mediates viral entry to the host cell by binding to the receptor, angiotensin-converting enzyme 2 (ACE2), and triggering the membrane fusion. The S protein consists of two functionally distinct subunits: S1 and S2. S1 is responsible for receptor binding, whereas S2 is responsible for membrane fusion. The N-terminal domain of S1 comprises the receptor-binding domain (RBD) that directly interacts with the ACE2 receptor. Most COVID-19 vaccines are based on the S protein or its RBD as the vaccine antigen [13,30,31].

In clinical trials evaluating the safety profile, the subunit vaccines performed well. The combined rate of local and systemic reactions was significantly lower for protein subunit vaccines (33.0%, 22.3%, respectively) than for other COVID-19 vaccines. Pain and tenderness at the injection site were the most common local reactions, and fatigue and headachess were the predominant systemic reactions [24,32,33]. Adverse events were relatively mild in the elderly, with both the frequency and intensity of local and systemic reactions decreasing with age. In clinical trials, the incidence of adverse events, including non-fatal serious adverse events and deaths, was similar between the vaccine and placebo groups. Reporting rates of common adverse events after mass public vaccination were lower than in clinical trials [32,34].

Currently, 19 protein-based subunit COVID-19 vaccines have been approved worldwide (https://www.who.int/publications/m/item/draft-landscape-of-covid-19-candidate-vaccines, accessed date 10 March 2023).

The vaccine developed by Novavax under the trade name Nuvaxovid (also referred to as NVX-CoV2373) is the most widely approved protein subunit COVID-19 vaccine. It is approved in 40 countries: Australia, Austria, Belgium, Bulgaria, Canada, Croatia, Cyprus, Czechia, Denmark, Estonia, Finland, France, Germany, Greece, Hungary, Iceland, Ireland, Israel, Italy, Latvia, Liechtenstein, Lithuania, Luxembourg, Malta, the Netherlands, New Zealand, Norway, Poland, Portugal, the Republic of Korea, Romania, Singapore, Slovakia, Slovenia, Spain, Sweden, Switzerland, Taiwan, the United Kingdom of Great Britain and Northern Ireland, and the United States of America. The vaccine has been evaluated in 22 clinical trials in 14 countries (https://www.who.int/publications/m/item/draft-landscape-of-covid-19-candidate-vaccines, accessed date 10 March 2023). Nuvaxovid contains a version of the S protein found in the original SARS-CoV-2 virus strain (Wuhan-Hu-1) and produced by recombinant DNA technology using a baculovirus expression system in an insect cell line that is derived from Sf9 cells of the *Spodoptera frugiperda* species. The vaccine also contains polysorbate 80 (PS80) stabilizing nanoparticles and a patented saponin-based adjuvant, Matrix-M™. Matrix-M™ consists of Fraction-A and Fraction-C of *Quillaja saponaria* Molina extract (https://www.ema.europa.eu/en/documents/product-information/nuvaxovid-epar-product-information_en.pdf, accessed date 28 July 2023). The vaccine is intended for primary and booster vaccinations to prevent the coronavirus disease in people aged 12 years and older.

The efficacy of the vaccine was 90.4% in clinical trials conducted in the United States and Mexico and 89.7% in clinical trials conducted in the United Kingdom. The original strain of SARS-CoV-2 and variants of concern such as Alpha, Beta, and Delta were the most common viral strains in circulation when the studies were ongoing. However, there are limited data concerning the vaccine efficacy against Omicron subvariants (https://www.ema.europa.eu/en/medicines/human/EPAR/nuvaxovid, accessed date 20 March 2023).

Additional studies also showed an increase in antibody levels when a booster dose of Nuvaxovid was given to adults after primary vaccination with Nuvaxovid, mRNA vaccines, or adenovirus vector vaccines (https://www.ema.europa.eu/en/human-regulatory/overview/public-health-threats/coronavirus-disease-covid-19/covid-19-public-health-emergency-international-concern-2020-23/covid-19-latest-updates-archive, accessed date 28 July 2023). Nuvaxovid is also manufactured by the Serum Institute of India under the trade name Covovax. Covovax is approved in six countries: Bangladesh, India, Indonesia, the Philippines, South Africa, and Thailand. The vaccine has been evaluated in seven clinical trials conducted in three countries (India, Indonesia, and South Africa) (https://www.who.int/publications/m/item/draft-landscape-of-covid-19-candidate-vaccines, accessed date 10 March 2023). Moreover, the same formulation as the Novavax vaccine was used in Takeda under the trade name TAK-019, which has only been approved in Japan (https://www.who.int/publications/m/item/draft-landscape-of-covid-19-candidate-vaccines, accessed date 10 March 2023). 

Novavax formulation vaccines are generally well tolerated. The most commonly observed post-vaccination adverse reactions of local and systemic nature included tenderness and pain at the injection site, fatigue, myalgia, headachess, arthralgia, general malaise and nausea, and vomiting. The adverse reactions were usually mild or moderate and were resolved within a few days. They occurred more frequently after the second dose.

Another protein subunit COVID-19 vaccine is VidPrevtyn Beta developed by Sanofi/GSK. It is approved in 30 countries: Austria, Belgium, Bulgaria, Croatia, Cyprus, Czechia, Denmark, Estonia, Finland, France, Germany, Greece, Hungary, Iceland, Ireland, Italy, Latvia, Liechtenstein, Lithuania, Luxembourg, Malta, the Netherlands, Norway, Poland, Portugal, Romania, Slovakia, Slovenia, Spain, and Sweden. The vaccine has been evaluated in three clinical trials in two countries (USA, France) (https://www.who.int/publications/m/item/draft-landscape-of-covid-19-candidate-vaccines, accessed date 10 March 2023). VidPrevtyn Beta, similarly to Nuvaxovid, contains SARS-CoV-2 spike protein produced with the recombinant DNA technology using a baculovirus expression system in an insect cell line that is derived from Sf9 cells of the fall armyworm, *Spodoptera frugiperda.* However, the protein S used in the vaccine comes from the Beta variant (B.1.351 strain) of the virus. As an adjuvant, AS03 composed of squalene, DL-α-tocopherol, and polysorbate 80 is used. VidPrevtyn Beta is indicated only as a booster in adults who previously received the adenoviral vector COVID-19 vaccine or mRNA vaccine (https://www.ema.europa.eu/en/documents/product-information/vidprevtyn-beta-epar-product-information_en.pdf, accessed date 28 July 2023). The clinical studies with VidPrevtyn Beta were carried out by comparing the immune response elicited by this new vaccine with that elicited by an approved comparator vaccine, i.e., mRNA vaccine Comirnaty (Pfizer/BioNTech). The studies showed that the VidPrevtyn Beta booster should be at least as effective as the mRNA vaccine, including against the SARS-CoV-2 Omicron BA.1 subvariant (https://www.ema.europa.eu/en/documents/product-information/vidprevtyn-beta-epar-product-information_en.pdf, accessed date 28 July 2023).

Research at the Vaccine Center for Genetic Engineering and Biotechnology (CIGB) in Cuba developed a vaccine with the trade name Abdala. Abdala is a recombinant protein subunit vaccine that uses the yeast *Pichia pastoris* as the microorganism for expression. The vaccine formulation includes the recombinant protein receptor binding domain (RBD) of SARS-CoV-2 and aluminum hydroxide gel as an adjuvant [35]. This vaccine successfully passed five clinical trials and is currently approved in six countries: Cuba, Mexico, Nicaragua, Saint Vincent and the Grenadines, (the Bolivarian Republic of) Venezuela, and Vietnam (https://www.who.int/publications/m/item/draft-landscape-of-covid-19-candidate-vaccines, accessed date 10 March 2023). Another vaccine developed in Cuba at the Vaccine Instituto Finlay de Vacunas is Soberana 02, which is approved in four countries: Cuba, (the Islamic Republic of) Iran, Nicaragua, and (the Bolivarian Republic of) Venezuela (https://www.who.int/publications/m/item/draft-landscape-of-covid-19-candidate-vaccines, accessed date 10 March 2023). Soberana 02 is based on SARS-CoV-2 RBD produced in genetically modified CHO cells and conjugated to tetanus toxoid (TT). Aluminum hydroxide is used as an adjuvant [36]. The safety and efficiency of the vaccine were evaluated in seven clinical trials conducted in Cuba and Iran. As a booster, Soberana Plus has been approved, which contains dimeric-RBD as an antigen. Soberana Plus is approved in two countries: Cuba and Belarus. In this case, the clinical trial was carried out only in Cuba (https://www.who.int/publications/m/item/draft-landscape-of-covid-19-candidate-vaccines, accessed date 10 March 2023). 

The vaccine with the trade name MVC-COV1901 developed by Medigen is a subunit vaccine based on the stable prefusion spike protein (S-2P) of SARS-CoV-2 and is adjuvanted with CpG 1018 an adjuvant and aluminum hydroxide [37]. After 15 performed trials (phase I–III), it was approved in four countries: Eswatini, Paraguay, Somaliland, and Taiwan (https://www.who.int/publications/m/item/draft-landscape-of-covid-19-candidate-vaccines, accessed date 10 March 2023). The vaccine developed by Anhui Zhifei Longcom under the trade name Zifivax is approved in four countries: China, Colombia, Indonesia, and Uzbekistan. It contains a dimeric form of the RBD and aluminum hydroxide as an adjuvant [38].

The EpiVacCorona vaccine developed by the Vector State Research Center of Virology and Biotechnology contains a composition of chemically synthesized peptide immunogens of the S protein of SARS-CoV-2 conjugated to a carrier protein and adsorbed on aluminum hydroxide. The carrier protein is a recombinant SARS-CoV-2 protein N [39]. The vaccine has been approved in Cambodia, Russian Federation, Turkmenistan, and Venezuela (https://www.who.int/publications/m/item/draft-landscape-of-covid-19-candidate-vaccines, accessed date 10 March 2023). 

The Corbevax vaccine developed by Biological E Limited has been approved in two countries: India and Botswana (https://www.who.int/publications/m/item/draft-landscape-of-covid-19-candidate-vaccines, accessed date 10 March 2023). The vaccine contains the receptor binding domain (RBD) of the S protein of SARS-CoV-2 and is adjuvanted with CpG1018 and aluminum hydroxide [40].

Protein subunit vaccines approved only in one country include the Noora Vaccine developed by the Bagheiat-allah University of Medical Sciences, Razi Cov Pars developed by the Razi Vaccine and Serum Research Institute, and SpikoGen developed by Vaxine/CinnaGen Co., all three of which are approved in Iran, Recombinant SARS-CoV-2 Vaccine (CHO Cell) developed by the National Vaccine and Serum Institute and approved in the United Arab Emirates, IndoVac developed by Bio Farma and approved in Indonesia, Aurora-CoV developed by the Vector State Research Center of Virology and Biotechnology and approved in the Russian Federation, SKYCovione (GBP510) developed by SK Bioscience Co. Ltd. and approved in the Republic of Korea, and V-01 developed by Livzon Mabpharm Inc. and approved in China (https://www.who.int/publications/m/item/draft-landscape-of-covid-19-candidate-vaccines, accessed date 10 March 2023). In March 2023, the Bimervax vaccine (also referred to as COVID-19 Vaccine HIPRA and PHH-1V) was approved in the European Union by the EMA as a booster in individuals aged 16 and older who previously received an mRNA COVID-19 vaccine. The vaccine was developed by the Spanish company Hipra Human Health S.L. The vaccine contains SARS-CoV-2 virus recombinant spike (S) protein receptor binding domain (RBD) fusion heterodimer originated from B.1.351 and B.1.1.7 strains and was produced by recombinant DNA technology using a plasmid expression vector in a CHO cell line. The vaccine is adjuvanted with SQBA. The SQBA adjuvant contains squalene, polysorbate 80, sorbitan trioleate, sodium citrate, and citric acid (https://www.ema.europa.eu/en/documents/product-information/bimervax-epar-product-information_en.pdf, accessed date 28 July 2023). In the clinical trial, the vaccine was slightly less reactogenic than the mRNA vaccine Comirnaty. The most frequent adverse events were as follows: injection site pain, fatigue, and headaches. The effectiveness of the vaccine was verified and confirmed by measuring the level of neutralizing antibodies for the ancestral Wuhan-Hu-1 strain, the Beta variant, the Delta variant, and the Omicron BA.1 variant. The vaccine efficacy against severe COVID-19 was similar to the Comirnaty vaccine [41].

### 2.3. mRNA Vaccines

The mRNA vaccines are part of an innovative approach by delivering a nucleotide sequence encoding the antigen or antigens selected for their high potential to induce a protective immune response. COVID-19 mRNA vaccines are the first vaccines based on the mRNA technology that have been approved for use in public health programs, although studies of this technology have been conducted for several years and a few potential vaccines against other infectious diseases (e.g., CMV, Zika, influenza, rabies, and malaria) have reached the second phase of testing. The vaccines contain messenger RNA (mRNA) encoding the antigen of interest, i.e., S protein in the case of COVID-19 vaccines. The S protein is the main surface protein used to bind to a receptor. Previously, during the development process of vaccines against MERS and SARS, it was observed that protein complex S is required for membrane fusion and host cell infection [42,43]. To deliver RNA to the cells, various approaches have been utilized including polymer-based nanoparticle formulation, lipid encapsulation, and incorporation of 5′-cap or 3′poly-A sequences protamine condensation, etc. Immediately after the vaccination, the RNA internalizes to cells and protein translation starts using the cell machinery to produce the antigen encoded in the injected RNA molecules [6].

It should also be highlighted that another type of RNA-based vaccine is studied, namely self-amplifying RNA (seRNA). The seRNA vaccines contain viral positive-stranded RNA. The seRNA, apart from the antigen, also encodes the factors necessary for the amplification of RNA within the target cell, like RNA polymerase [6].

The mRNA vaccines have a lot of advantages, e.g., high potency, ability to initiate protein production without the need for nuclear entry, capacity for rapid development, potential for low-cost manufacture, and safe administration using non-viral delivery and they do not require growing the virus at the laboratory. The desired antigen or multiple antigens can be expressed from mRNA without the need to adjust the production process, which offers maximum flexibility and efficiency in development [33] (Sanofi, 2020).

To the present day, mRNA COVID-19 vaccines developed by four companies have been approved in one country at least.

Comirnaty is an mRNA vaccine against COVID-19 (contained in lipid nanoparticles), which was developed by Pfizer (USA) and BioNTech (Germany) (also referred to as Tozinameran, BNT162b2). It has been approved in 149 countries: Albania, Angola, Antigua and Barbuda, Argentina, Armenia, Australia, Austria, Azerbaijan, Bahamas, Bahrain, Bangladesh, Barbados, Belgium, Belize, Benin, Bermuda, Bhutan, (the Plurinational State of) Bolivia, Bosnia and Herzegovina, Botswana, Brazil, Brunei Darussalam, Bulgaria, Burkina Faso, Cabo Verde, Cameroon, Canada, Chile, Colombia, Cook Islands, Costa Rica, Croatia, Cyprus, Czechia, Côte d’Ivoire, Democratic Republic of the Congo, Denmark, Dominican Republic, Ecuador, Egypt, El Salvador, Estonia, Eswatini, Ethiopia, Faroe Islands, Finland, France, Gabon, Georgia, Germany, Ghana, Greece, Greenland, Grenada, Guatemala, Guinea, Guyana, Honduras, Hongkong, Hungary, Iceland, Indonesia, Iraq, Ireland, Israel, Italy, Jamaica, Japan, Jordan, Kazakhstan, Kenya, Kosovo, Kuwait, Kyrgyzstan, Lao People’s Democratic Republic, Latvia, Lebanon, Libya, Liechtenstein, Lithuania, Luxembourg, Malawi, Malaysia, the Maldives, Malta, Mauritius, Mexico, Monaco, Mongolia, Montenegro, Morocco, Namibia, Nepal, the Netherlands, New Zealand, Nicaragua, Nigeria, Niue, North Macedonia, Norway, Oman, Pakistan, Panama, Papua New Guinea, Paraguay, Peru, the Philippines, Poland, Portugal, Puerto Rico, Qatar, the Republic of Korea, Republic of Moldova, Romania, Rwanda, Saint Kitts and Nevis, Saint Lucia, Saint Vincent and the Grenadines, Saudi Arabia, Serbia, Singapore, Slovakia, Slovenia, South Africa, Spain, Sri Lanka, Sudan, Sweden, Switzerland, Taiwan, Thailand, Timor-Leste, Togo, Tokelau, Tonga, Trinidad and Tobago, Tunisia, Turkey, Uganda, Ukraine, the United Arab Emirates, the United Kingdom of Great Britain and Northern Ireland, the United Republic of Tanzania, the United States of America, Uruguay, the Vatican, Vietnam, West Bank, and Zambia. The vaccine has been evaluated in 100 clinical trials in 39 countries (https://www.who.int/publications/m/item/draft-landscape-of-covid-19-candidate-vaccines, accessed date 10 March 2023). It contains single-stranded, 5’ capped messenger RNA (mRNA) produced by cell-free in vitro transcription on a DNA template, encoding the spike protein (S) of the SARS-CoV-2 virus. The most common side effects observed were injection site pain, fatigue, headaches, muscle pain and chills, joint pain, and fever, as well as swelling of the injection site (https://www.ema.europa.eu/en/documents/product-information/bimervax-epar-product-information_en.pdf, accessed date 28 July 2023). The vaccine efficacy has been estimated at 94.6% (95% CI: 89.9–97.3) in its protection against COVID-19 disease in adults and adolescents from 16 years of age, as well as 100% effectiveness in its protection against severe COVID-19 requiring hospitalization and death due to COVID-19 [44]. However, the efficacy has decreased with the emergence and domination of the new SARS-CoV-2 variants [45]. Therefore, the company developed a modified vaccine that contains mRNA specific for new variants of the virus: Omicron BA.1 and Omicron BA.4/BA.5.

Comirnaty Bivalent Original/Omicron BA.1 (also referred to as BNT162b2 (B.1.1.529), BNT162b2 Bivalent (WT/OMI BA.1), riltozinameran, and tozinameran) has been approved in 35 countries: Australia, Austria, Belgium, Bulgaria, Canada, Croatia, Cyprus, Czechia, Denmark, Estonia, Finland, France, Germany, Greece, Hungary, Iceland, Ireland, Italy, Japan, Latvia, Liechtenstein, Lithuania, Luxembourg, Malta, the Netherlands, Norway, Poland, Portugal, the Republic of Korea, Romania, Slovakia, Slovenia, Spain, Sweden, and the United Kingdom of Great Britain and Northern Ireland. The vaccine has been evaluated in three clinical trials in five countries (https://www.who.int/publications/m/item/draft-landscape-of-covid-19-candidate-vaccines, accessed date 10 March 2023). It contains molecules called mRNA, which have instructions for making the spike proteins of the original SARS-CoV-2 and the Omicron subvariant BA.1. This vaccine is administered as a booster dose. The most common side effects observed were injection site pain, fatigue, headaches, muscle pain and chills, joint pain, fever, and swelling of the injection site. Vaccine efficacy was calculated at 81.8% (two-dose booster immunization) against symptomatic disease and 93% for preventing severe COVID-19 and death [17].

Comirnaty Bivalent Original/Omicron BA.4/BA.5 (also referred to as BNT162b2 Bivalent (WT/OMI BA.4/BA.5), tozinameran, and famtozinameran) has been approved in 33 countries: Austria, Belgium, Bulgaria, Canada, Croatia, Cyprus, Czechia, Denmark, Estonia, Finland, France, Germany, Greece, Hungary, Iceland, Ireland, Italy, Japan, Latvia, Liechtenstein, Lithuania, Luxembourg, Malta, the Netherlands, Norway, Poland, Portugal, Romania, Slovakia, Slovenia, Spain, Sweden, and the United States of America. The vaccine has been evaluated in four clinical trials in one country (https://www.who.int/publications/m/item/draft-landscape-of-covid-19-candidate-vaccines, accessed date 10 March 2023). It contains molecules called mRNA, which have instructions for making the spike proteins of the original strain of SARS-CoV-2 and the Omicron subvariants BA.4 and BA.5 (EMA, accessed date 30 March 2023). This vaccine is also utilized as a booster dose. The most common side effects observed were injection site pain, fatigue, headaches, muscle pain and chills, joint pain, fever, and swelling of the injection site. The vaccine efficacy was calculated at 94.6% against symptomatic disease (https://www.ema.europa.eu/en/documents/product-information/bimervax-epar-product-information_en.pdf, accessed date 28 July 2023).

Spikevax is an mRNA vaccine against COVID-19 developed by Moderna (also referred to as mRNA-1273, elasomeran). It has been approved in 88 countries: Argentina, Australia, Austria, Bangladesh, Belgium, Bhutan, Botswana, Brunei Darussalam, Bulgaria, Canada, Chile, Colombia, Croatia, Cyprus, Czechia, Democratic Republic of the Congo, Denmark, Egypt, Estonia, Faroe Islands, Fiji, Finland, France, Germany, Ghana, Greece, Greenland, Guatemala, Guyana, Haiti, Honduras, Hungary, Iceland, India, Indonesia, Ireland, Israel, Italy, Kenya, Kuwait, Latvia, Libya, Liechtenstein, Lithuania, Luxembourg, Malawi, Malaysia, Maldives, Malta, Mexico, (Federated States of) Micronesia, Mongolia, Nepal, the Netherlands, Nigeria, Norway, Pakistan, Paraguay, Philippines, Poland, Portugal, Puerto Rico, Qatar, the Republic of Korea, Republic of Moldova, Romania, Rwanda, Saint Vincent and the Grenadines, Saudi Arabia, Seychelles, Singapore, Slovakia, Slovenia, Spain, Sri Lanka, Suriname, Sweden, Switzerland, Taiwan, Thailand, Trinidad and Tobago, Uganda, Ukraine, the United Arab Emirates, the United Kingdom of Great Britain and Northern Ireland, the United States of America, Vietnam, and West Bank. The vaccine has been evaluated in 70 clinical trials in 24 countries (https://www.who.int/publications/m/item/draft-landscape-of-covid-19-candidate-vaccines, accessed date 10 March 2023). The vaccine contains single-stranded, 5’ capped messenger RNA (mRNA) produced through cell-free in vitro transcription on a suitable DNA template, encoding the spike protein (S) of the SARS CoV-2 virus named elasomeran. The mRNA is encapsulated within lipid nanoparticles (LNPs) SM-102. The most common side effects observed were injection site pain, fatigue, headaches, muscle pain, joint pain, chills, nausea/vomiting, armpit swelling/tenderness, fever, and injection site swelling and redness. The vaccine efficacy was calculated at 94.1% in protecting against COVID-19 symptoms (95% CI: 89.3–96.8) and 90.9% effectiveness in patients at risk of severe COVID-19 with co-morbidities and in immunocompromised patients [46,47].

Similar to the Comirnaty vaccine, Spikevax has also been developed for the new variants of SARS-CoV-2. Spikevax Bivalent Original/Omicron BA.1 (also referred to as mRNA-1273.214) has been approved in 38 countries: Australia, Austria, Belgium, Bulgaria, Canada, Croatia, Cyprus, Czechia, Denmark, Estonia, Finland, France, Germany, Greece, Hungary, Iceland, Ireland, Italy, Japan, Latvia, Liechtenstein, Lithuania, Luxembourg, Malta, the Netherlands, Norway, Poland, Portugal, the Republic of Korea, Romania, Singapore, Slovakia, Slovenia, Spain, Sweden, Switzerland, Taiwan, and the United Kingdom of Great Britain and Northern Ireland. The vaccine has been evaluated in five clinical trials in four countries (https://www.who.int/publications/m/item/draft-landscape-of-covid-19-candidate-vaccines, accessed date 10 March 2023). It contains mRNA molecules that have instructions for forming the spike proteins of the original SARS-CoV-2 (elasomeran) and the Omicron subvariant BA.1 (imelasomeran) (https://www.ema.europa.eu/en/medicines/human/EPAR/spikevax, accessed date 30 March 2023). This vaccine is administered as a booster dose. The most common side effects observed included injection site pain, fatigue, headaches, muscle pain, joint pain, chills, nausea/vomiting, armpit swelling/tenderness, fever, and injection site swelling and redness. Vaccine efficacy was calculated at 81.8% (two-dose booster immunization) against symptomatic disease and 93% for preventing severe COVID-19 and death [13].

Spikevax Bivalent Original/Omicron BA.4/BA.5 (also referred to as mRNA-1273.222) has been approved in 33 countries: Austria, Belgium, Bulgaria, Canada, Croatia, Cyprus, Czechia, Denmark, Estonia, Finland, France, Germany, Greece, Hungary, Iceland, Ireland, Italy, Japan, Latvia, Liechtenstein, Lithuania, Luxembourg, Malta, the Netherlands, Norway, Poland, Portugal, Romania, Slovakia, Slovenia, Spain, Sweden, and the United States of America. It has been evaluated in two clinical trials in one country (https://www.who.int/publications/m/item/draft-landscape-of-covid-19-candidate-vaccines, accessed date 10 March 2023). The vaccine contains mRNA molecules that have instructions for making the spike proteins of the original SARS-CoV-2 (elasomeran) and the Omicron subvariant BA.4 and BA.5 (dawesomeran) (https://www.ema.europa.eu/en/medicines/human/EPAR/spikevax, accessed date 30 March 2023). This vaccine is used as a booster dose. The most common side effects observed were injection site pain, fatigue, headaches, muscle pain, joint pain, chills, nausea/vomiting, armpit swelling/tenderness, fever, and injection site swelling and redness. The vaccine efficacy is much higher when administered as a booster and protects against symptomatic and severe disease (in particular, hospitalizations and deaths) (https://www.ema.europa.eu/en/medicines/human/EPAR/spikevax, accessed date 30 March 2023).

Comirnaty (BNT162b2) and Spikevax (mRNA-1273) vaccines had similar effectiveness against the wild-type variant of SARS-CoV-2 (Alpha). But the effectiveness dropped to 64.8% and 65.0%, respectively, against the Delta variant and 31.6% and 25.6%, respectively, against the Omicron variant. The application of booster doses increased the effectiveness for a short period of time [48]. Therefore, the modified vaccines that contain mRNA specific for new variants are crucial for keeping immune protection.

The early studies of BA.4/5- and BA.1-based bivalent vaccines’ (Moderna or Pfizer/BioNTech) effectiveness showed very good protection against hospitalization and death from COVID-19 for several months (74% for BA.1 vaccines and 61.8–80.5% for BA.4/5 vaccines) during the Omicron-domination period. The protection against symptomatic infection was estimated at 65% for BA.1 vaccines and 76% for BA.4/5 vaccines.

An immunogenicity study revealed that individuals that received boosters of bivalent BA.4/5 vaccines developed 2.91 to 6.3 higher neutralizing antibody titers to the BA.4/5 Omicron subvariant (including newer BQ.1.1 and XBB.1) than those who received the original formulation. Participants who received the bivalent BA.1 formulation had 1.6 to 1.7 times higher neutralizing antibody titers against the Omicron BA.1 variant than people who received the original formulation (Australian Government. COVID-19 vaccine information, accessed date 10 August 2023).

GEMCOVAC-19 (also referred to as Gemcovac), developed by Gennova Biopharmaceuticals Limited, has been approved in India, where it has been evaluated in two clinical trials. The vaccine contains self-amplifying mRNA, which has instructions for making the spike protein (S-protein) of the original SARS-CoV-2. The GEMCOVAC-19 vaccine does not require ultra-low temperature storage (https://www.who.int/publications/m/item/draft-landscape-of-covid-19-candidate-vaccines, accessed date 10 March 2023). 

TAK-919 is manufactured by Takeda. This vaccine has the same formulation as the Spikevax (mRNA-1273) developed by Moderna and has been approved in Japan. The vaccine has been evaluated in two clinical trials in Japan (https://www.who.int/publications/m/item/draft-landscape-of-covid-19-candidate-vaccines, accessed date 10 March 2023). The most common side effects observed were headaches, fatigue, myalgia, arthralgia, nausea/vomiting, chills, and fever (NIH, Clinical Trials).

The AWcorna vaccine developed by Walvax Biotechnology, Suzhou Abogen Biosciences, and the PLA Academy of Military Science has been approved in Indonesia. The vaccine has been evaluated in four clinical trials in three (https://www.who.int/publications/m/item/draft-landscape-of-covid-19-candidate-vaccines, accessed date 10 March 2023). It contains mRNA encoding the receptor-binding domain (RBD) of spike glycoprotein (S protein) of SARS-CoV-2. The mRNA is produced using the in vitro transcription method according to the DNA template and is then encapsulated in lipid nanoparticles. The most common side effects observed were fever, headaches, fatigue/asthenia, myalgia arthralgia, nausea, and chills (Walvax: 2 April 2023). The vaccine efficacy was 83.58% against the wild-type coronavirus strains considered common, but its efficacy dropped to 71.17% against the Omicron variant (https://www.reuters.com/business/healthcare-pharmaceuticals/indonesia-drug-agency-approves-chinas-walvax-mrna-vaccine-emergency-use-2022-09-29/, accessed date 2 April 2023).

### 2.4. Vector-Based Vaccines (VVr, VVnr)

Vector vaccines, similar to mRNA vaccines, are the latest generation of vaccines. They contain a modified viral vector into which a gene encoding an antigen is introduced, in this case, the S spike protein from SARS-CoV-2. The viral vector is the carrier of the antigen. There are currently two types of viral vector vaccines: non-replicating and self-replicating or replicating. Following administration of the vaccine, the antigen is expressed, and a strong humoral and cellular immune response is initiated, which is similar to that occurring in natural infection. Cytokines, chemokines, and costimulatory molecules are secreted to provide an adjuvant effect. This is particularly true for replication-competent vector vaccines, where new viral entities are formed providing enhanced antigen presentation that elicits strong responses. In this way, the administration of a smaller dose of a replicating vector vaccine already gives a sufficient effect and allows obtaining vaccines with a better safety and efficacy profile. In the case of replication-deficient viral vectors, there may be a potential reduction in long-term effectiveness, which is due to the response induction mechanism, where each virus particle used as a vector in the case of non-replication vaccines is able to infect one host cell using the built-in transgene and produce appropriate vaccine antigens; however, it is unable to create new viral units after administration of the vaccine. More doses of the vaccine in the primary vaccination and/or booster doses are needed to obtain a durable and strong immune response [24,49,50].

The best-known and most characterized are adenoviruses. Their genome consists of early genes (E1-4) encoding proteins that initiate and maintain replication and late genes (L1-5) encoding structural proteins. Deletion of the E1A and E1B genes results in a replication-deficient virus. The E3 gene, which encodes proteins that inhibit the ability of infected cells to respond to the immune system and eliminate the virus, is often removed. These genes are replaced by the actual gene encoding the vaccine antigen [50,51].

The advantages of vector-based adenovirus vaccines also include the following: (i) unlike other viral vectors (e.g., based on lentiviruses), they do not interfere with the host genome and the viral DNA remains episomal; (ii) deletion of the E1 and E3 genes allows insertion of large sequences foreign genes; and (iii) high specificity of gene delivery [50,51].

The disadvantage of vaccines based on human adenovirus (Ad5 serotype) may be that as a virus circulating in the population, it can lead to the development of immunity to this type of virus by limiting the expression of the transgene. Therefore, in the development of vaccines, atypical serotypes were used: Ad26 and Ad35 and non-human adenoviral vectors such as chimpanzee (ChAdY25, Ad36, Ad68) and gorilla (GRAd32) [24,51].

Researchers’ interest in using viral vectors in COVID-19 vaccine development stemmed from their previous use in gene therapy and in developing vaccine platforms for AIDS, malaria, Zika, and Ebola vaccine candidates [52,53]. Currently, nine vector-based COVID-19 vaccines have been approved worldwide. All contain non-replicating viral vectors three (https://www.who.int/publications/m/item/draft-landscape-of-covid-19-candidate-vaccines, accessed date 10 March 2023). 

The Jcovden vaccine developed by Janssen (Johnson & Johnson) (also referred to as Ad26.COV2.S, Ad26COVS1, and JNJ-78436735) has been approved in 113 countries: Afghanistan, Antigua and Barbuda, Australia, Austria, Bahamas, Bahrain, Bangladesh, Barbados, Belgium, Belize, Benin, (the Plurinational State of) Bolivia, Botswana, Brazil, Bulgaria, Burkina Faso, Burundi, Cambodia, Cameroon, Canada, Central African Republic, Chile, Colombia, Croatia, Cyprus, Czechia, Côte d’Ivoire, Democratic Republic of the Congo, Denmark, Djibouti, Egypt, Estonia, Eswatini, Ethiopia, Faroe Islands, Finland, France, Gabon, Gambia, Germany, Ghana, Greece, Guinea, Guinea-Bissau, Hungary, Iceland, India, Indonesia, (the Islamic Republic of) Iran, Ireland, Italy, Jamaica, Kenya, Kuwait, Lao People’s Democratic Republic, Latvia, Lesotho, Liberia, Libya, Liechtenstein, Lithuania, Luxembourg, Madagascar, Malawi, Malaysia, the Maldives, Mali, Malta, Mauritania, Mauritius, Mexico, (Federated States of) Micronesia, Morocco, Namibia, Nepal, the Netherlands, New Zealand, Nigeria, Norway, Papua New Guinea, Peru, the Philippines, Poland, Portugal, Puerto Rico, the Republic of Korea, Republic of Moldova, Romania, Rwanda, Saint Lucia, Saint Vincent and the Grenadines, Saudi Arabia, Senegal, Slovakia, Slovenia, South Africa, South Sudan, Spain, Sudan, Sweden, Switzerland, Thailand, Togo, Trinidad and Tobago, Tunisia, Uganda, Ukraine, the United Kingdom of Great Britain and Northern Ireland, the United Republic of Tanzania, the United States of America, Vietnam, Zambia, and Zimbabwe. The vaccine has been evaluated in 26 clinical trials in 25 countries three (https://www.who.int/publications/m/item/draft-landscape-of-covid-19-candidate-vaccines, accessed date 10 March 2023). It contains adenovirus type 26 encoding the SARS-CoV-2 spike glycoprotein. The Ad26 is produced in the proprietary PER.C6 TetR cell line derived from human embryonic retinal tissue and by recombinant DNA technology. The company used the AdVac platform previously utilized in the production of the Ebola vaccine (https://www.ema.europa.eu/en/medicines/human/EPAR/mvabea, accessed date 27 July 2023), [54]. The common adverse events included fatigue, headaches, myalgia, nausea, and fever above 38.0 °C. Most side effects were of mild or moderate nature, occurred within 1–2 days after vaccination, and were of short duration (1–2 days). The vaccine efficacy in preventing COVID-19 disease in adults was estimated at 66.9% after a single dose in adults 14 days after vaccination and 66.1% after a single dose in adults 28 days after vaccination. The effectiveness of the vaccine against severe COVID-19 disease was estimated at 76.7% and 85.4%, respectively (main analysis). In updated analyses, the vaccine efficacy was estimated at 76.1% against symptomatic and severe COVID-19 beyond 14 days after vaccination (https://www.ema.europa.eu/en/medicines/human/EPAR/jcovden-previously-covid-19-vaccine-janssen, accessed date 10 August 2023).

In clinical trials evaluating the relevance of booster vaccination, the efficacy in boosting immunity after primary vaccination was confirmed. An increase in neutralizing and protein S-binding antibodies has been shown following a booster dose 2 months or more apart from the primary vaccination course. In addition, JCOVDEN was shown to increase antibody responses in people vaccinated with two doses of Spikevax and two doses of Comirnaty. Similar results were obtained with the combination of a primary vaccination with the Vaxzevria vaccine and a booster vaccination with the JCOVDEN vaccine (https://www.ema.europa.eu/en/medicines/human/EPAR/jcovden-previously-covid-19-vaccine-janssen, accessed date 10 August 2023).

The Vaxzevria vaccine developed by Oxford/AstraZeneca (also referred to as AZD1222, ChAdOx1 nCoV-19) has been approved in 149 countries: Albania, Algeria, Angola, Antigua and Barbuda, Argentina, Armenia, Australia, Austria, Azerbaijan, Bahamas, Bangladesh, Barbados, Belgium, Belize, Benin, Bermuda, Bosnia and Herzegovina, Botswana, Brazil, Brunei Darussalam, Bulgaria, Burkina Faso, Cambodia, Cameroon, Canada, Central African Republic, Chile, Colombia, Costa Rica, Croatia, Cyprus, Czechia, Côte d’Ivoire, Democratic Republic of the Congo, Djibouti, Dominican Republic, Ecuador, Egypt, El Salvador, Estonia, Eswatini, Ethiopia, Fiji, Finland, France, Gambia, Georgia, Germany, Ghana, Greece, Grenada, Guatemala, Guinea, Guinea-Bissau, Guyana, Haiti, Hungary, Iceland, India, Indonesia, (the Islamic Republic of) Iran, Iraq, Ireland, Italy, Jamaica, Japan, Jordan, Kenya, Kiribati, Kosovo, Kuwait, Kyrgyzstan, Latvia, Lebanon, Lesotho, Liberia, Libya, Liechtenstein, Lithuania, Luxembourg, Madagascar, Malawi, Malaysia, the Maldives, Mali, Malta, Mauritania, Mauritius, Mexico, Mongolia, Montenegro, Morocco, Mozambique, Nauru, Nepal, Netherlands, New Zealand, Nicaragua, Niger, Nigeria, North Macedonia, Oman, Pakistan, Panama, Papua New Guinea, Paraguay, Peru, Philippines, Poland, Portugal, the Republic of Korea, Republic of Moldova, Romania, Rwanda, Saint Kitts and Nevis, Saint Vincent and the Grenadines, Samoa, Sao Tome and Principe, Saudi Arabia, Senegal, Serbia, Sierra Leone, Slovakia, Slovenia, Solomon Islands, South Sudan, Spain, Sri Lanka, Sudan, Sweden, Taiwan, Tajikistan, Thailand, Timor-Leste, Togo, Trinidad and Tobago, Tunisia, Tuvalu, Uganda Ukraine, the United Arab Emirates, the United Kingdom of Great Britain and Northern Ireland, Uruguay, Uzbekistan, Vanuatu, Vietnam, West Bank, Yemen, and Zambia. The vaccine has been evaluated in 73 clinical trials in 34 countries (https://www.who.int/publications/m/item/draft-landscape-of-covid-19-candidate-vaccines, accessed date 10 March 2023). It contains chimpanzee adenovirus encoding the SARS-CoV-2 Spike glycoprotein (ChAdOx1-S) produced in genetically modified human embryonic kidney (HEK) 293 cells derived from a fetus aborted in 1972 and by recombinant DNA technology [55] (https://www.ema.europa.eu/en/medicines/human/EPAR/vaxzevria, accessed date 10 August 2023). The most common adverse events were pain/tenderness at the injection site, fatigue, myalgia, malaise, fever ≥ 38 °C, chills, arthralgia, and nausea. Most of the side effects were of mild or moderate nature and occurred a few days after the vaccination. Very rare cases of thrombosis with thrombocytopenia syndrome were reported in post-marketing experience in the first three weeks after the vaccination. After the second dose, side effects were milder and occurred less often. The vaccine efficacy was estimated at 60% based on studies from the UK (COV002 study) and Brazil (COV003 study). Another study conducted in the United States, Peru, and Chile, with 21% of participants over the age of 65, found a 74% reduction in the number of cases of symptomatic COVID-19 in those who received two doses of the vaccine (second dose 4 weeks after the first injection) compared to those given a control injection (https://www.ema.europa.eu/en/medicines/human/EPAR/vaxzevria, accessed date 10 August 2023), (https://www.who.int/publications/m/item/draft-landscape-of-covid-19-candidate-vaccines, accessed date 10 March 2023). 

Further, phase II and phase III clinical trials evaluated the efficacy of a Vaxzevria vector booster in subjects who had previously received a primary course of Vaxzevria. Non-lower pseudo-neutralizing antibody titers against the parental strain were demonstrated compared to antibody titers elicited by the 2-dose primary series. At the same time, an increase in the humoral response was also noted in people who had previously received their primary vaccination with the mRNA vaccine against COVID-19. On this basis, the SPC of both vaccines allows the use of heterologous COVID-19 vaccination courses. (https://www.ema.europa.eu/en/medicines/human/EPAR/vaxzevria, accessed date 10 August 2023).

The Covishield vaccine manufactured by the Serum Institute of India contains the Oxford/AstraZeneca vaccine formulation (also referred to as ChAdOx1 nCoV-19). The World Health Organization (WHO) states that ChAdOx1-S (recombinant) products (AstraZeneca AZD1222 and SII COVISHIELD^TM^) are considered equivalent ( https://www.seruminstitute.com/health_faq_covishield.php#references(accessed date 10 August 2023). It has been approved in 49 countries: Afghanistan, Antigua and Barbuda, Argentina, Bahamas, Bahrain, Bangladesh, Barbados, Belize, Bhutan, (the Plurinational State of) Bolivia, Botswana, Brazil, Cabo Verde, Canada, Côte d’Ivoire, Dominica, Egypt, Ethiopia, Ghana, Grenada, Guyana, Honduras, Hungary, India, Jamaica, Lebanon, Madagascar, the Maldives, Morocco, Myanmar, Namibia, Nepal, Nicaragua, Nigeria, Republic of Moldova, Saint Kitts and Nevis, Saint Lucia, Saint Vincent and the Grenadines, Seychelles, Solomon Islands, Somalia, South Africa, Sri Lanka, Suriname, Syrian Arab Republic, Togo, Tonga, Trinidad and Tobago, and Ukraine. The vaccine has been evaluated in 6 clinical trials in 1 country [WHO: 2 December 2022]. The composition of the Covishield vaccine is the same as in the Vaxzevria vaccine. The most common adverse events included injection site tenderness and pain, headaches and fatigue. Most side effects were of mild or moderate nature and usually resolved within a few days after the vaccination. The clinical trials were conducted in India, among medical personnel [56].

The Convidecia vaccine developed by the Chinese company CanSino Biologics (also referred to as Ad5-nCoV) has been approved in 10 countries: Argentina, Chile, China, Ecuador, Hungary, Indonesia, Malaysia, Mexico, Pakistan, and the Republic of Moldova. The vaccine has been evaluated in 14 clinical trials in 6 countries https://www.who.int/publications/m/item/draft-landscape-of-covid-19-candidate-vaccines (accessed date 30 March 2023). It contains viral particles (vp) of human type 5 adenovirus encoding SARS-CoV-2 Spike (S) glycoprotein and excipients https://www.who.int/publications/i/item/WHO-2019-nCoV-vaccines-SAGE-recommendation-Ad5-nCoV-Convidecia-background (accessed date 19 May 2022). All noticed side effects were mild or moderate and lasted no longer than 48 h. No serious reactions were reported within 28 days after the vaccination. The most common side effects observed were high fever, fatigue, joint pain, headaches, muscle ache and pain at the injection site (54%). Elevated levels of bilirubin (8%), alanine aminotransferase (9%) and fasting blood sugar (4%) have been observed in a few patients 7 days after the vaccination. The vaccine efficacy was estimated as 65.7% against symptomatic and 90.98% against severe COVID-19 infection [57].

The Convidecia Air vaccine is the second vaccine developed by CanSino Biologics (also referred to asAd5-nCoV-IH). This Chinese manufacturer used the same adenoviral vector-based technology platform as in the Convidecia intramuscular vaccine. Convidecia Air is an oral aerosol vaccine for use as a booster. It has been approved in 2 countries: China and Morocco. The vaccine has been evaluated in 5 clinical trials in 4 countries https://www.who.int/publications/m/item/draft-landscape-of-covid-19-candidate-vaccines (accessed date 30 March 2023). It contains a recombinant, replication-defective human Ad5 vector encoding the full-length spike protein of the wild-type SARS-CoV-2 virus, Wuhan-Hu-1 [32]. The most common adverse events observed were fever, fatigue, and headaches and occurred less frequently than after intramuscular vaccination. The safety analysis was performed after administration of two aerosol doses at the interval of 28 days. The vaccine efficacy was observed by assessing the immunogenicity of two doses of aerosolized Ad5-nCoV: a single dose can elicit a strong cellular response and two doses produce SARS-CoV-2 neutralizing antibodies at a similar level as one dose of this vaccine solution for injection [32].

The Russian Gamaleya National Center of Epidemiology and Microbiology developed the Sputnik V vaccine (also referred to as Gam-COVID-Vac). The vaccine has been approved in 74 countries: Albania, Algeria, Angola, Antigua and Barbuda, Argentina, Armenia, Azerbaijan, Bahrain, Bangladesh, Belarus, (the Plurinational State of) Bolivia, Bosnia and Herzegovina Brazil, Cambodia, Cameroon, Chile, Djibouti, Ecuador, Egypt, Gabon, Ghana, Guatemala, Guinea, Guyana, Honduras, Hungary, India, Indonesia, (the Islamic Republic of) Iran, Iraq, Jordan, Kazakhstan, Kenya, Kyrgyzstan, Lao People’s Democratic Republic, Lebanon, Libya, the Maldives, Mali, Mauritius, Mexico, Mongolia, Montenegro, Morocco, Myanmar, Namibia, Nepal, Nicaragua, Nigeria, North Macedonia, Oman, Pakistan, Panama, Paraguay, Philippines, Republic of Moldova, Republic of the Congo, Russian Federation, Rwanda, Saint Vincent and the Grenadines, San Marino, Serbia, Seychelles, Sri Lanka, Syrian Arab Republic, Tunisia, Turkey, Turkmenistan, the United Arab Emirates, Uzbekistan, (the Bolivarian Republic of) Venezuela, Vietnam, West Bank, and Zimbabwe. The vaccine has been evaluated in 25 clinical trials in eight countries https://www.who.int/publications/m/item/draft-landscape-of-covid-19-candidate-vaccines (accessed date 30 March 2023). It contains two types of adenovirus, rAd type 26 (rAd26) and rAd type 5 (rAd5), both of which have an embedded SARS-CoV-2 full-length S glycoprotein gene. The vaccine is administered in two doses and, unlike other vector-based vaccines, the first dose uses the Ad26 adenovirus and the second dose contains the Ad5 adenovirus. This is to overcome the pre-existing immunity to adenoviruses in the population [58,59]. The most common adverse events were injection site reactions, headaches, asthenia, and flu-like symptoms and most of them were mild (94.0%). Serious adverse events were reported during the study (0.3% in subjects who received the vaccine and 0.4% in those in the placebo group). None of those were considered as related to the vaccination [59]. The vaccine efficacy was estimated at 91.6% (phase III interim results as a vaccine under the name Gam-COVID-Vac) 21 days after the first dose of the vaccine. The effectiveness of the vaccine in preventing more severe forms of the disease was estimated at 100% [58,59,60].

Another two vaccines developed by the Gamaleya National Center of Epidemiology and Microbiology are Gam-COVID-Vac and Sputnik Light. The Gam-COVID-Vac vaccine (also referred to as Sputnik, rAd5) has been approved in one country: the Russian Federation. The vaccine has been evaluated in two trials: one phase I and one phase II https://www.who.int/publications/m/item/draft-landscape-of-covid-19-candidate-vaccines (accessed date 30 March 2023).

The Sputnik Light vaccine has been approved in 26 countries: Angola, Argentina, Armenia, Bahrain, Belarus, Benin, Cambodia, Egypt, India, (the Islamic Republic of) Iran, Kazakhstan, Kyrgyzstan, Lao People’s Democratic Republic, Mauritius, Mongolia, Nicaragua, Philippines, Republic of the Congo, Russian Federation, San Marino, Tunisia, Turkmenistan, the United Arab Emirates, the United Republic of Tanzania, (the Bolivarian Republic of) Venezuela, and West Bank. The vaccine has been evaluated in seven clinical trials in three countries https://www.who.int/publications/m/item/draft-landscape-of-covid-19-candidate-vaccines (accessed date 30 March 2023). It contains the first component of the two-dose vaccine “Sputnik V”, recombinant human adenovirus serotype number 26 (rAd26). The most common adverse events included mild systemic and local reactions at the injection site. No serious adverse events were detected. The single-dose vaccine Sputnik Light has been observed to induce strong cellular and humoral immune responses [61].

The iNCOVACC vaccine developed by Bharat Biotech (also referred to as BBV154) has been evaluated as part of four clinical trials in one country and was approved in one country: India https://www.who.int/publications/m/item/draft-landscape-of-covid-19-candidate-vaccines (accessed date 30 March 2023). The vaccine contains a viral vector ChAd36-SARS-CoV-2-S chimpanzee adenovirus. The iNCOVACC vaccine is the second intranasal vaccine against COVID-19 in the world in a primary series and heterologous booster that is administered at the site of entry of the SARS-CoV-2 virus and induces not only a humoral and cellular response but also a strong mucosal (sIgA) response [62].

### 2.5. DNA Vaccines

Similar to mRNA vaccines, the DNA vaccines had not been proven before COVID-19 pandemic. The mechanism of action of DNA vaccines is based on the direct introduction into appropriate tissues of a plasmid containing the DNA sequence encoding the antigens, which are expressed and translated into proteins to evoke the immune response. The use of plasmid DNA in vaccines brings some benefits over traditional approaches, including stimulation of both cellular and humoral immunity, high vaccine stability, absence of any infectious agents, and relative ease of large-scale production. Compared to live attenuated vaccines, DNA vaccines do not involve the risks arising from the potential attenuation of vaccine strains [63]. However, the disadvantage of DNA vaccines is that they are limited to protein immunogens [64]. Scientists are also developing technologies that help DNA to enter cells and transfer them to specific cells, or which can act as aids in stimulating or directing immune responses. The first vaccines of this type licensed for marketing are likely to use plasmid DNA derived from bacterial cells. The WHO has developed guidelines for controlling DNA vaccines to provide a scientifically sound basis for the production of DNA vaccines, for human use, and to ensure their continued safety and efficacy. Individual countries can flexibly use the said guidelines to develop their national guidelines for DNA vaccines https://www.who.int/teams/health-product-and-policy-standards/standards-and-specifications/vaccines-quality/dna (accessed date 26 January 2023). Currently, only one DNA vaccine against COVID-19 has been approved globally, namely, ZyCoV-D, developed by the Indian pharmaceutical company Zydus Cadila and approved in India. ZyCoV-D contains DNA plasmids pVAX1, which encode the spike protein of SARS-CoV-2 and IgE signal peptide together with a promoter sequence. The spike gene region was selected from Wuhan Hu1 isolate [65,66]. In the nuclei of cells, plasmids are converted into mRNA, which travels to the main body of the cell and is translated into the spike protein itself. The body’s immune system produces immune cells. Plasmids typically degrade, but immunity is preserved [65]. The efficacy of the ZyCoV-D vaccine estimated in clinical trials was 66.6%. The vaccine was proven to be 100% efficacious to prevent severe cases of COVID-19 after two doses. The majority of observed adverse events were mild to moderate [66].

### 2.6. Virus-Like Particle (VLP) Vaccines

Virus-like particles (VLPs) are nanostructures that imitate viruses but without the potential to replicate due to a lack of viral genomic material. They are based on capsid proteins for self-assembly, offering a flexible platform for interactions between proteins inside host cells [67]. VLPs self-organization is a natural process. Highly organized, ranging from 20 to 200 nm, they may possess various geometric structure forms of icosahedral, helical symmetry, rod shape structure, or globular in shape [68]. Associations between them are facilitated by thermodynamic equilibrium based on “van der Waals” forces, hydrogen bonds, and hydrophobic and electrostatic interactions during the nucleation and growth phases [69]. VLPs are most often obtained with genetic engineering methods, thanks to which it is also possible to obtain new types of particles, such as a mosaic composed of several proteins or with fragments of various proteins attached [70]. They can mimic the general structure of virus particles while maintaining full immunogenicity. It is believed that they may be considered a cheaper and easier alternative to vaccines characterized by safety and high immunogenicity. The mechanism of action of these particles is complicated, but they constitute a unique molecular pattern associated with the pathogen (PAMP) [71]. VLPs can contain several antigens in their structure, stimulating immune cells. They are also specific to the type of tissue penetrated [72]. Moreover, the VLPs have high adjuvant abilities. They may present T and B cell epitopes inducing humoral, cellular, and mucosal immune responses. VLP production proceeds in three steps: construction and cloning of viral structural genes, their expression in a heterologous environment, separation, and purification, and, finally, the verification process. VLPs can be produced in prokaryotic and eukaryotic models, dividing them into a non-enveloped VLP (hepatitis E virus, human papillomavirus) [73,74], an enveloped VLP (influenza virus and hepatitis virus), and a chimeric VLP (against non-infectious diseases, such as hypertension, Alzheimer’s, nicotine addiction, allergies, and diabetes) [68,75,76]. VLPs can be developed and produced using different types of expression systems like bacteria, yeasts, mammalian cells, insect cells, and plants [77].

Currently, there is only one VLP vaccine against COVID-19 approved in one country, namely the COVIFENZ COVID-19 vaccine developed by Medicago Inc. and approved in Canada. The vaccine contains plant-based virus-like particles of SARS-CoV-2 spike (S) protein (original strain) with an AS03 adjuvant (manufactured by GlaxoSmithKline). The plant-derived vaccine platform used the transfection of *Nicotiana benthamiana*, an Australian plant, and disarmed *Agrobacterium tumefaciens* in the delivery of foreign episomal DNA to the plant cell nucleus. The ability to express the envelope protein of a human virus in plants and create envelope VLPs that can grow from cell membranes was demonstrated for the first time by [78]. In that system, protein expression is not expressed by the plasmid. The *Agrobacterium* enzyme cuts the plasmid fragment and transfers it to the plant cell. Following the S protein expression process, self-assembled VLPs were isolated from the matrix of the plant and purified by filtration and chromatography [79]. The “plant system” used in the production of COVIFENZ has many advantages, such as rapid production, high scalability, low cost, safety, and the capacity to produce multimeric proteins. Moreover, plants are free of endotoxins, oncogenes, and mammalian pathogens. Therefore, they are safe and exempt from the costs of refinement and product screening.

Based on trial participants aged 18 to 64, the COVIFENZ COVID-19 vaccine exhibits an effectiveness of 71% in protecting against COVID-19. Possible side effects can last for a few hours to a few days after the vaccination and include redness, soreness, and swelling at the injection site as well as more general symptoms, such as chills, fatigue, joint pain, headaches, mild fever, muscle pain, nasal congestion, sore throat, cough, nausea, diarrhea, and generally feeling unwell (malaise) (https://www.canada.ca/en.html, accessed date 4 March 2023).

## 3. Vaccines Quality and Safety

The procedure for marketing authorization of a medicinal product, including a vaccine, is a very complex and time-consuming process. Long-term tests conducted as part of preclinical and clinical trials at the centers responsible for them are necessary. Initially, preclinical tests involve numerous laboratory tests, mainly on an animal model. Clinical trials are allowed only after their positive outcome. As a standard, it can be divided into three basic stages from which the data obtained are intended to confirm or exclude the safety and effectiveness of the preparations being developed. They must also ensure that the benefits of their use significantly outweigh the side effects. In the first stage of the research, the safety profile of the tested product, its impact on living organisms, pharmacokinetics, toxicity, and possible interactions of the preparation’s ingredients with other substances are analyzed. These tests are carried out on a group of several dozen healthy volunteers.

The obtained results allow proceeding to the second stage to be carried out on a much higher number of volunteers. The second stage of research is aimed at confirming the effectiveness of the preparation and determining the dosing schedule. The research takes into account the patient’s age, sex, and health condition, and all research samples are verified with a placebo group. This is the most important stage of clinical trials; it helps to confirm the advantages of using the test drug. Once the first two stages are concluded with positive results, the third stage proceeds, which is the longest and at the same time very important because it is used to assess the long-term effects of medicinal products, confirms the results obtained in the previous stages, and allows to finally evaluate the appropriateness of use of the tested preparations [80] (https://www.ema.europa.eu/en/ich-e6-r2-good-clinical-practice-scientific-guideline#revision-1-section, accessed date 28 July 2023).

However, to combat the COVID-19 pandemic, the acceleration in vaccine development was crucial. It was possible due to the previous research on the development of vaccine candidates against related coronaviruses, like SARS-CoV and MERS-CoV. For example, the virus antigen that induces the immune response and protection was known thanks to the earliest studies. Moreover, it is possible to combine the phases of clinical trials. For instance, phase I and II can be conducted together or phase II and III can be combined [81].

The vaccination of vulnerable populations against COVID-19 has been a priority. Therefore, for the marked acceleration of vaccine development and licensing procedures, the authorities have implemented special procedures, such as the “rolling review”, which is a regulatory tool that the European Medicines Agency (EMA) uses to speed up the assessment of data for medicine or a vaccine during a public health emergency. In the conventional mode of action, the evaluation of medicine needs a maximum of 210 active days (without clock stops necessary for supplementing the documentation by a Marketing Authorization Holder (MAH)). The rolling review mode enables reducing the review timelines to < 150 working days. The process consists of several review cycles, where each cycle is pre-agreed between the applicant and the EMA, and questions from previous cycles must be addressed before the next cycle starts [82].

After the approval, the safety and effectiveness of COVID-19 vaccines are continuously monitored using real-world data. The monitoring includes analysis of suspected side effects reported by patients, parents, and healthcare professionals, observational studies, and ongoing clinical studies, etc. The aim of this monitoring is to identify and evaluate any new information that arises promptly, including any safety signals that are relevant to the benefit–risk balance of these vaccines. In case of identification of any reasonable possibility that a vaccine could have caused a suspected side effect, this information is immediately included in the vaccine’s product information, which is available for patients and healthcare professionals (EMA, Monitoring of COVID-19 medicines). For the monitoring and surveillance of COVID-19 vaccines’ safety and efficacy, specific guidelines and recommendations have been developed by public health agencies such as the WHO, EMA, CDC, and FDA (for example, the Pharmacovigilance Plan of the EU Regulatory Network for COVID-19 Vaccines (europa.eu), COVID-19 vaccines: safety surveillance manual (who.int), and COVID-19 Vaccine Safety Surveillance (FDA), accessed date 10 August 2023). Moreover, the emerging of new SARS-CoV-2 variants is monitored as well as their impact on vaccine efficacy. Special guidelines for variant strains update to COVID-19 vaccines have also been developed (for example, the Procedural guidance for variant strain(s) update to vaccines intended for protection against human coronavirus (EMA), accessed date 10 August 2023).

All vaccines are very carefully controlled not only by the manufacturer but also by independent laboratories for medicines. The WHO recommends an independent batch release by National Control Laboratories to provide assurance of the quality, safety, and efficacy of vaccines. COVID-19 vaccines are no exception. In parallel with the development of COVID-19 vaccines, the National Control Laboratories for Medicines were preparing for quality testing of the new vaccines. Regulatory authorities around the globe worked on establishing control requirements to ensure the safety and efficacy of the new COVID-19 vaccines. The rapid development and authorization of various types of new vaccines required equally rapid preparation for the independent control of the vaccines. Stringent control of the quality of each vaccine lot is extremally important since the vaccine could be applied to millions of people during the mass vaccination campaigns on all continents of the globe [81,83].

## 4. Vaccinated Population

Achieving global vaccine coverage was one of the obstacles during the COVID-19 pandemic. Together with policy responses and restrictions, vaccination effort was employed against the causative severe acute respiratory syndrome coronavirus 2 (SARS-CoV-2) [84]. As of June of 2023, thirteen vaccines against COVID-19 have been placed on the Emergency Use Listing by the World Health Organization (WHO EUL) (https://extranet.who.int/pqweb/vaccines/vaccinescovid-19-vaccine-eul-issued, accessed date 28 July 2023). Many of them have been also approved by appropriate regulatory authorities authorized in the USA, Canada, European Union, Africa, and the Caribbean (Table 2).

For children, the risk of infection and severe disease from SARS-CoV-2 is low [85]. The majority of vaccines are approved for use in adults aged 18 and above [86]. Based on the positive evaluation of the available safety and efficacy data, a number of vaccines are now also being authorized for use in children. In accordance with the WHO publication, as of August of 2022, the Pfizer-BioNTech BNT162b2 and Moderna mRNA-1273 vaccines were granted the authorization for emergency use in some countries to be used in the age groups of 6 months and above.

The EUA recommends vaccination in 12–17-year-olds in all EU/EEA countries. The European Medicines Agency (EMA) has approved vaccination for 6–11-year-olds with Spikevax and Cominarty. Sinovac-CoronaVac and BBIBP-CorV were approved by Chinese authorities for the age indication of 3–17 years, and Covaxin was approved in India for the age indication of 12–17 years but has not yet received the WHO EUL for this age indication.

Based on the current knowledge, pregnancy does not seem to correlate with a high risk of becoming infected with COVID-19. However pregnant and recently pregnant people are at higher risk for severe illness from COVID-19 once infected. COVID-19 during pregnancy has also been associated with an increased risk of complications and the likelihood of preterm birth [1] (https://www.who.int/news-room/questions-and-answers/item/coronavirus-disease-covid-19-pregnancy-and-childbirth, accessed date 28 July 2023). The CDC recommends a COVID-19 vaccination for all people aged 6 months and older. This includes people who are pregnant, breastfeeding, currently trying to get pregnant, or those who might become pregnant in the future (https://www.cdc.gov/coronavirus/2019-ncov/vaccines/recommendations/pregnancy.html, accessed date 28 July 2023).

In accordance with the European Centre for Disease Prevention and Control (https://www.ema.europa.eu/en/medicines/human/summaries-opinion/comirnaty-1, accessed date 30 March 2023), the COMIRNATY vaccine has been approved in 149 countries and over 684 million doses have been administered in EU/UEA counties. The second most administered vaccine has been SpikeVax, which was administered in over 195 million doses in EU/UEA counties.

Vaccine policies greatly influenced the total number of administered vaccines. As of December 2022, almost all countries, excluding Yemen, Senegal, and Sierra Leone, implemented policies for universal vaccine administration (https://ourworldindata.org/coronavirus, accessed date 14 December 2022).

While comparing the number of vaccines administered in different countries, the number of doses administered per 100 people and the share of the population that has completed the initial vaccination protocol are being taken into consideration as there are discrepancies in the number of vaccine boosters recommended for each vaccine and differences in the countries’ population [84].

The number of administered doses shown in Figure 3 distinguishes countries like Cuba (388.54 doses), Chile (319.78 doses), and Japan (307.46 doses), where doses administered per 100 people are above 300, as the chart includes boosters (https://ourworldindata.org/coronavirus, accessed date 14 December 2022). The USA’s and most of the EU countries’ vaccination levels per 100 people are within the range of 200–250.

If we compare the percentage of the population that completed the initial vaccination protocol (two doses for most vaccines and one or three for several manufacturers) shown in Figure 4, the countries with the highest rating are Qatar (105.8%) and Chile (90.3%). Canada, Australia, New Zeeland, and most European countries are within the range of 70–80% of the population.

Data for the cumulative uptake (%) of the primary course by age group shows that in EU/EEA countries (data as of February of 2023), the highest level of vaccination (including boosters) was in the 60+ group. In countries such as Iceland and Ireland, the said age group was fully vaccinated (100%), with Denmark, Portugal, and Belgium being near that status (<98%) (https://qap.ecdc.europa.eu/public/extensions/COVID-19/vaccine-tracker.html#uptake-tab, accessed date 16 February 2023). A similar trend can be observed in most countries like the USA (95%) https://data.cdc.gov/widgets/gxj9-t96f?mobile_redirect=true, accessed date 28 July 2023), Japan (92%), and Israel (92.8%) (Our World in Data). The opposite trend can be noticed in children aged below 9. They have the lowest cumulative uptake of vaccination (≤30%). In EU/EEA countries, the highest vaccination rate for the age group 0–4 was in Austria (1.8%), and for the age group 5–9, it was in Portugal (33.1%), Spain (32.2%), and Denmark (29.9%) (https://qap.ecdc.europa.eu/public/extensions/COVID-19/vaccine-tracker.html#uptake-tab, accessed date 16 February 2023). In Israel and Japan, the vaccination rates for the age group 5–11 were the lowest: 25% (https://ourworldindata.org/coronavirus, accessed date 16 February 2023). In the USA, the vaccination rate in those age groups was not included in the data source (https://data.cdc.gov/widgets/gxj9-t96f?mobile_redirect=true, accessed date 28 July 2023).

Outside of government policies, achieving global vaccine coverage during the COVID-19 pandemic was also challenged by matters of production, affordability, allocation, and deployment of vaccines [87]. Vaccine hesitancy and concerns regarding its safety also affected vaccine impact. Consideration of mentioned obstacles is especially important for vaccine roll-out in low- and middle-income countries (LMICs) [88]. Most LMIC countries have been relying on the COVID-19 Vaccines Global Access (COVAX, accessed date 16 August 2023) facility to obtain vaccines, which aims to provide counties with doses for 20% of their population. During the pandemic, LIMC countries experienced additional challenges in vaccine distribution, compared to high-income countries (HIC), despite their significant experience from the Expanded Programme on Immunization (EPI, accessed date 16 August 2023). Problems that were present in HICs countries were governments’ limited funds to buy vaccines, vaccine shortages, few trained service providers, suboptimal cold-chain systems, fear of adverse events, and rumors of vaccine inefficacy, which remain issues for COVID-19 vaccination as LMICs build on their EPI to deliver COVID-19 vaccines [88].

## 5. Perspectives and Conclusions

The acceleration of the COVID-19 vaccine process of development and approval by competent bodies enables the production and distribution of millions of doses of vaccines worldwide. Mass vaccination helps limit the deaths caused by SARS-CoV-2. The initial concerns that vaccines would only be available to selected countries were quickly resolved. However, it has to be highlighted that the quality and effectiveness of COVID-19 vaccines used around the world may vary, especially in the context of the emergence of new variants of SARS-CoV-2. Among 50 vaccines approved by at least one country, 13 have been approved by the WHO. Moreover, the safety of all the approved vaccines is continuously monitored to ensure that any possible risks are detected and managed as early as possible. It must be kept in mind that vaccines’ immunogenicity (including those against COVID-19) is affected by various factors. It was shown that the following factors might contribute to different vaccine responses in different regions: (i) age—infants do not have a mature immune system and therefore their cell-mediated immune responses are very weak, aging may lead to immune-depression and inflammations and a high prevalence of comorbidities, and in aged patients, immunogenicity is lower; (ii) poor diet and malnutrition and composition of gut microflora—immune responses to vaccination is weaker in low- and middle-income countries in comparison to high-income ones; (iii) it was shown that humoral responses after vaccination against COVID-19 were weaker in obese people; (iv) immunogenicity also varies between males and females (women usually have higher immune responses), which can lead to differences in responses to vaccine; and (v) some studies suggest that host genetics also may contribute to vaccination outcomes [89]. Table 3 shows a comparison of inactivated vaccines’ efficacy as an example.

The new vaccine platforms that have been approved for the first time for human application present a great potential for future development. For example, the possible application of new mRNA-based therapeutics includes not only new vaccines against pathogens other than SARS-CoV-2 but also therapies for cancer, genetic disorders, metabolic disorders, cardiovascular and vascular disorders, and Alzheimer’s disease [50]. The top 20 disease classes covered in mRNA patents are shown in Figure 5. The increase in mRNA platform application is reflected by an increase in the number of scientific publications and patent applications (Figure 6) [90].

In terms of infectious diseases, the speed of mRNA vaccines’ development and production as well as their flexibility and adaptability to new variants of pathogens increases the chance to stop the spread of the changing pathogens escaping immune response by rapid adaptation of a vaccine.

Adenovirus vectors might be used not only for vaccine development but also in cancer and gene therapies [84,85]. Currently, adenovirus-based gene therapy clinical trials account for 50% of total worldwide gene therapy trials [91]. Nevertheless, adenovirus platforms have been mainly applied for the development of novel vaccines and cancer therapies. Some of the adenovirus cancer therapies are currently in the third phase of clinical trials [92]. VLPs might have applications in targeted drug delivery systems and for use in gene therapy [74]. Recently, the VLPs’ properties have been investigated for their application in a theranostic platform, particularly for cancer [92].

The COVID-19 pandemic forced an accelerated development of vaccines, but it has also triggered rapid drug and therapy development for non-infectious diseases. It might be assumed that the COVID-19 pandemic has had a great impact on the future of microbiology, medicine, and public health.

## Figures and Tables

**Figure 1 viruses-15-01786-f001:**
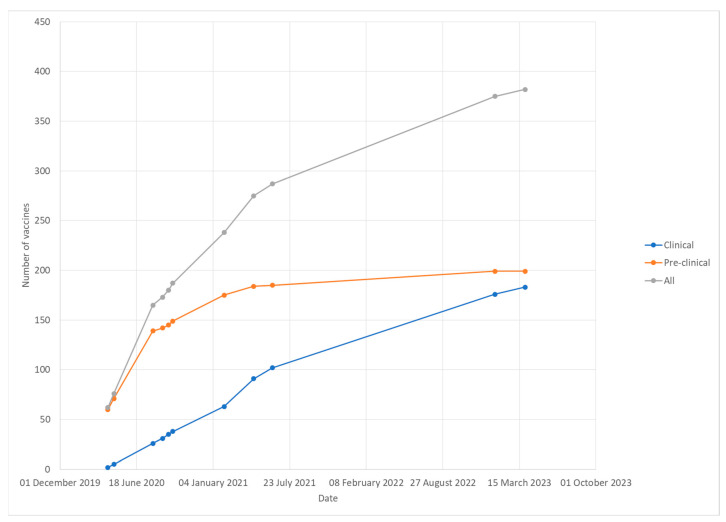
COVID-19 vaccine development dynamics from the beginning of the pandemic.

**Figure 2 viruses-15-01786-f002:**
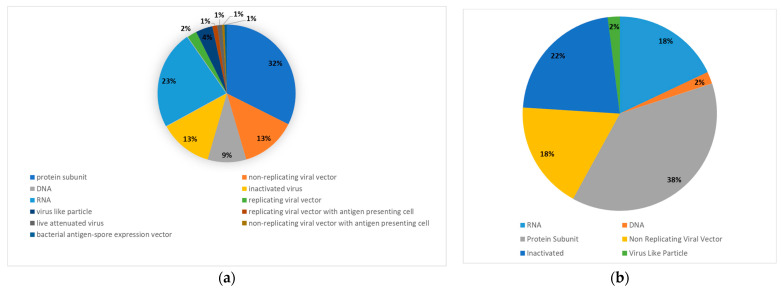
(**a**) The number of candidate vaccines and vaccine platforms; (**b**) the vaccines approved in at least in one country.

**Figure 3 viruses-15-01786-f003:**
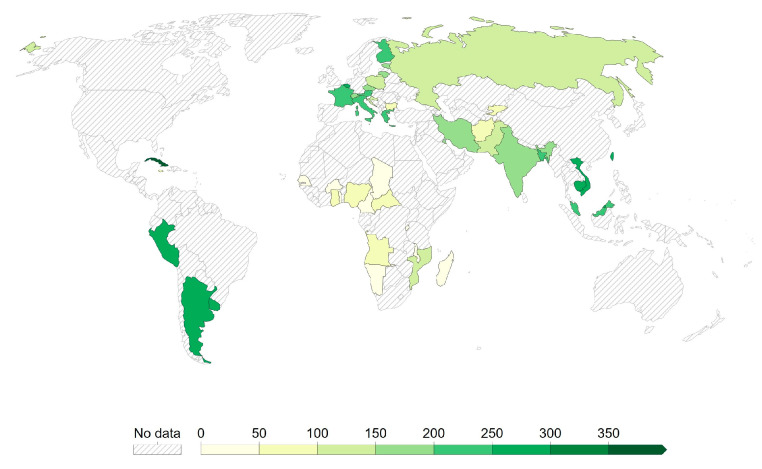
Total COVID-19 vaccine doses administered per 100 people (https://ourworldindata.org/coronavirus—last updated 28 July 2023).

**Figure 4 viruses-15-01786-f004:**
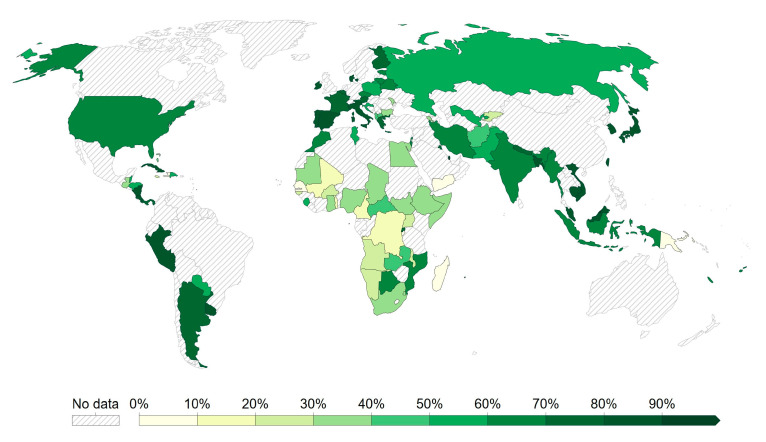
Share of people who completed the initial COVID-19 vaccination protocol. (https://ourworldindata.org/—last updated 28 July 2023).

**Figure 5 viruses-15-01786-f005:**
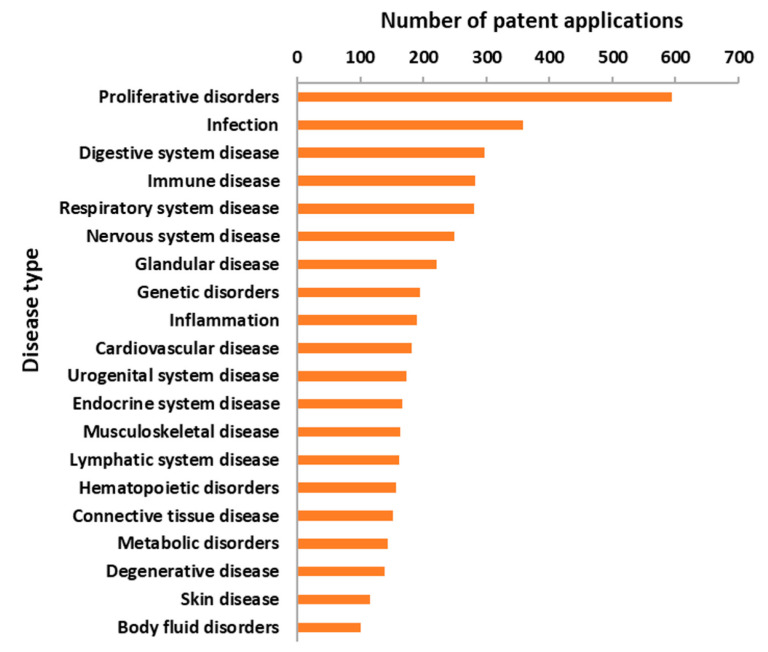
Number of mRNA patents for the top 20 disease types, according to Li et al. [90].

**Figure 6 viruses-15-01786-f006:**
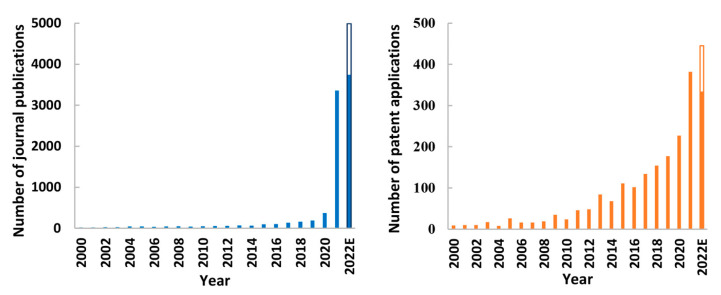
The annual number of published journal articles (**left**) and patents (**right**) on mRNA therapeutics and vaccines, according to Li et al. [90]. The data for 2022 include extrapolated numbers for October to December 2022.

**Table 1 viruses-15-01786-t001:** Vaccines approved for emergency use by at least one country (according to WHO and EMA data in May 2023).

Vaccine Platform	Vaccine	Vaccine Developer/Manufacturer	Number of Countries Where a Vaccine Is Approved
RNA	Comirnaty (Tozinameran, BNT162b2)	Pfizer/BioNTech	149
	Spikevax (mRNA-1273, Elasomeran)	Moderna	88
	GEMCOVAC-19 (Gemcovac)	Gennova Biopharmaceuticals Limited	1
	Spikevax Bivalent Original/Omicron BA.1 (mRNA-1273.214)	Moderna	38
	Spikevax Bivalent Original/Omicron BA.4/BA.5 (mRNA-1273.222)	Moderna	33
	Comirnaty Bivalent Original/Omicron BA.1 (BNT162b2 (B.1.1.529), BNT162b2 Bivalent (WT/OMI BA.1))	Pfizer/BioNTech	35
	Comirnaty Bivalent Original/Omicron BA.4/BA.5 (BNT162b2 Bivalent (WT/OMI BA.4/BA.5))	Pfizer/BioNTech	33
	TAK-919 (Moderna formulation)	Takeda	1
	AWcorna (mRNA)	Walvax	1
DNA	ZyCoV-D	Zydus Cadila	1
Protein subunit	Nuvaxovid (NVX-CoV2373)	Novavax	40
	Covovax (Novavax formulation)	Serum Institute of India	6
	Zifivax (RBD-Dimer, ZF2001)	Anhui Zhifei Longcom	4
	Noora vaccine (COVID-19 Recombinant RBD Protein Vaccine)	Bagheiat-allah University of Medical Sciences	1
	Corbevax (BECOV2A)	Biological E Limited	2
	Abdala (CIGB-66)	Center for Genetic Engineering and Biotechnology (CIGB)	6
	Soberana 02 (FINLAY-FR-2, Pastu Covac, Pastocovac)	Instituto Finlay de Vacunas Cuba	4
	Soberana Plus (FINLAY-FR-1A)	Instituto Finlay de Vacunas Cuba	2
	V-01	Livzon Mabpharm Inc	1
	MVC-COV1901	Medigen	4
	Recombinant SARS-CoV-2 Vaccine (CHO Cell) (Recombinant COVID-19 Vaccine (CHO cell, NVSI-06-08))	National Vaccine and Serum Institute	1
	IndoVac (SARS-CoV-2 Protein Subunit Recombinant Vaccine Adjuvanted With Alum + CpG 1018)	PT Bio Farma	1
	Razi Cov Pars	Razi Vaccine and Serum Research Institute	1
	VidPrevtyn Beta (CoV2 preS dTM adjuvanted vaccine (B.1.351), SP/GSK subunit B.1.351 vaccine)	Sanofi/GSK	30
	SKYCovione (GBP510)	SK Bioscience Co. Ltd.	1
	TAK-019 (Novavax formulation)	Takeda	1
	SpikoGen (COVAX-19)	Vaxine/CinnaGen Co.	1
	Aurora-CoV (EpiVacCorona-N)	Vector State Research Center of Virology and Biotechnology	1
	EpiVacCorona	Vector State Research Center of Virology and Biotechnology	4
	Bimervax (COVID-19 Vaccine HIPRA, PHH-1V)	Hipra Human Health S.L.	27
Non replicating viral vector	Convidecia (Ad5-nCoV)	CanSino	10
	Convidecia Air (Ad5-nCoV-IH)	CanSino	2
	Jcovden (Ad26.COV2.S, Ad26COVS1, JNJ-78436735)	Janssen (Johnson & Johnson)	113
	Vaxzevria (AZD1222, ChAdOx1 nCoV-19)	Oxford/AstraZeneca	149
	Covishield (Oxford/AstraZeneca formulation)	Serum Institute of India	49
	iNCOVACC (BBV154)	Bharat Biotech	1
	Gam-COVID-Vac (Sputnik, rAd5)	Gamaleya	1
	Sputnik Light	Gamaleya	26
	Sputnik V (Gam-COVID-Vac)	Gamaleya	74
Inactivated	Covaxin (BBV152)	Bharat Biotech	14
	Covilo (BBIBP-CorV)	Sinopharm (Beijing)	93
	CoronaVac	Sinovac	56
	KoviVac	Chumakov Center	3
	Turkovac (ERUCOV-VAC)	Health Institutes of Turkey	1
	FAKHRAVAC (MIVAC)	Organization of Defensive Innovation and Research	1
	QazVac (QazCovid-in)	Research Institute for Biological Safety Problems (RIBSP)	2
	KCONVAC (SARS-CoV-2 Vaccine (Vero Cells), KconecaVac)	Shenzhen Kangtai Biological Products Co.	2
	COVIran Barekat (COVID-19 Inactivated Vaccine)	Shifa Pharmed Industrial Co.	1
	Inactivated (Vero Cells)	Sinopharm (Wuhan)	2
	VLA2001	Valneva	33
Virus Like Particle (VLP)	Covifenz (CoVLP, MT-2766, Plant-based VLP)	Medicago	1

**Table 2 viruses-15-01786-t002:** COVID-19 vaccines granted Emergency Use Listing (EUL) by the WHO and vaccines authorized in the USA, Canada, European Union, Africa, and the Caribbean by appropriate regulatory authorities.

Vaccine (Manufacturer)	Age Indication	Number of doses	Schedule	EUL by WHO	EU—EMA	Canada—Canada Health	USA—FDA	The Caribbean—CRS	African Union—ART.
Comirnaty (Pfizer/BioNTech)	≥5 years	Primary series: 2	Day 0 + 28,	+	+	+	+	+	-
		Booster: 1	booster at least 3 months after completion of the primary series						
Comirnaty Original/Omicron (Pfizer/BioNTech)	≥12 years	Booster: 1	At least 3 months after completion of the primary series	+	+	+	+	+	-
Spikevax (Moderna)	≥6 years	Primary series: 2	Day 0 + 28,	+	+	+	+	+	-
		Booster: 1	booster at least 3 months after completion of the primary series						
Nuvaxovid (Novavax)	≥12 years	2	Day 0 + 21	+	+	+	+	+	-
Covovax (Serum Institute of India)	≥12 years according to WHO;	2	Day 0 + 21	+	-	-	-	+	-
	≥7 according to the manufacturer								
Convidencia (CanSino)	18–59 years according to WHO;	1	-	+	-	-	-	+	-
	≥18 years according to the manufacturer								
Jcovden (Janssen)	≥18 years	Primary series: 1	booster at least 3 months after completion of the primary series	+	+	+	+	+	+
		Booster: 1							
Vaxzevria (Oxford/AstraZeneca)	≥18 years	2	Day 0 + between 28 and 84 day	+	+	+	-	+	+
Covishield (Serum Institute of India)	≥18 years	2	Day 0 + between 28 and 84	+	-	-	-	+	+
Covaxin (Bharat Biotech)	≥18 years	2	Day 0 + 28	+	-	-	-	+	-
Covilo (Sinopharm)	≥18 years according to WHO;	2	Day 0 + between 21 and 28	+	-	-	-	+	+
	≥6 according to the manufacturer								
CoronaVac (Sinovac)	≥18 years	2	Day 0 + 28	+	-	-	-	+	+
Covid-19 vaccine (Valneva)	18–50 years	2	Day 0 + 28	-	+	-	-	-	-
VidPrevtyn Beta (Sanofi Pasteur)	≥18 years	Booster: 1	At least 4 months after a previous COVID-19 vaccine	-	-	-	-	-	-
Covifenz (Medicago)	18–64 years	2	Day 0 + 21	-	-	+	-	-	-
Bimervax (HIPRA Human Health S.L.U.)	≥16 years	Booster: 1	At least 6 months after a previous COVID-19 vaccine	-	+	-	-	-	-
SKYCovione™ (GBP510) (SK Bioscience Co., Ltd.)	18–64 years	2	Day 0 + 28	+	-	-	-	-	-

Note: Signs “+” and “-” has been used in the table to represent vaccine obtaining/or not vaccine authorization by appropriate regulatory authorities.EMA—European Medicines Agency, FDA—Food and Drug Administration, CRS—The Caribbean Regulatory System, ART—Africa Regulatory Taskforce

**Table 3 viruses-15-01786-t003:** Comparison of inactivated vaccines’ efficacy.

Inactivated Vaccine	Efficacy
Covaxin (BBV152)	77.8% against symptomatic COVID-19 disease. Efficacy against severe symptomatic COVID-19 disease—93.4%.
Covilo (BBIBP-CorV)	72.8% for WIV04 and 78.1% for HB02 against the symptomatic coronavirus disease. 79% against severe disease or hospitalization.
CoronaVac	More than 50% in Brazil in protection against symptomatic and asymptomatic SARS-CoV-2 infection. 83.5% in Turkey.
KoviVac	Seroconversion rate of 86.9% in seronegative participants.
Turkovac (ERUCOV-VAC)	Seroconversion rates for anti-SARS-CoV-2 spike antibodies at 94.3%. The neutralization assay yielded positivity rates of 51.2%
FAKHRAVAC (MIVAC)	Increase in the neutralizing antibody titer two and four weeks after administration of the second dose—69.6% and 73.4% of the participants in the vaccinated group, respectively.
QazVac (QazCovid-in)	The preventive efficacy of vaccination reached 82.0%.
KCONVAC (SARS-CoV-2 Vaccine (Vero Cells), KconecaVac)	Level of induced neutralizing antibodies and antigen-specific CD8 T cells after booster vaccination suggests that the vaccine is effective against the following variants: B.1.617.2 (Delta), (Alpha), B.1.351 (Beta), B.1.526.2, B.1.618, B.1.1.529 (Omicron), B.1.1.7 and B.1.617.3.
COVIran Barekat (COVID-19 Inactivated Vaccine)	Preventing the intensive care unit admission 3 months after the second dose reached 70%. The maximum effectiveness of the COVIran Barekat vaccine in preventing death 4 months after the administration of the second dose was over 90%.
VLA2001	After two weeks, the levels of tested antibodies were as high as in people after infection with SARS-CoV-2. The incidence of neutralizing antibodies against Omicron was 87%.

## Data Availability

Not applicable.

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
