# Peer review of "COVID-19 Vaccines over Three Years after the Outbreak of the COVID-19 Epidemic"

_viruses, 2023, doi:10.3390/v15091786_

Round 1

Reviewer 1 Report

In the review ‘COVID-19 vaccines over three years after the outbreak of the COVID-19 epidemic’, Aleksandra Anna Zasada and coauthors provides an overview of the development of COVID-19 vaccines, including the various platforms used for vaccine development. They discuss the advantages and disadvantages of different vaccine platforms and provides information on the status of COVID-19 vaccine development and distribution around the world. Authors paid appropriate credit to ideas, concepts and data that have been published recently. The review article is well written, structured, and easy to follow. However, there are a few shortcomings with respect to references, and including them will improve the manuscript.

Many of the statements here are backed up with appropriate references, but some are missing. I discuss them below:

Line 74-83, Line 90-95: ‘Live attenuated vaccines have been successfully produced for decades to prevent the most dangerous diseases such as tuberculosis, measles, mumps, rubella, chickenpox or rotavirus infection. A live attenuated vaccine usually produces immunity against all proteins of the target pathogen and therefore might be more effective against various genetic variants of the pathogen. However, live vaccines carry a certain risk of causing active disease in particularly vulnerable people, i.e. those with compromised immune systems.’

Line 435-439: ‘COVID-19 mRNA vaccines are the first vaccines based on 435 the mRNA technology that have been approved for use in public health programs, although studies of this technology have been conducted for several years and a few potential vaccines against other infectious diseases (e.g. CMV, Zika, influenza, rabies, malaria) have reached the second phase of testing.’

Line 458: [33], [Sanofi, 2020].?

The perspectives and conclusions section need substantial revision focusing on the findings of this systematic review only. The authors need to expand upon the perspectives part more.

Author Response

Dear Reviewer,

Thank you for review of the manuscript and the helpful and important suggestions. Below we present answers for all your suggestions.

Kind regards,

Aleksandra Zasada

Reviewer 1:

Comments and Suggestions for Authors

In the review ‘COVID-19 vaccines over three years after the outbreak of the COVID-19 epidemic’, Aleksandra Anna Zasada and coauthors provides an overview of the development of COVID-19 vaccines, including the various platforms used for vaccine development. They discuss the advantages and disadvantages of different vaccine platforms and provides information on the status of COVID-19 vaccine development and distribution around the world. Authors paid appropriate credit to ideas, concepts and data that have been published recently. The review article is well written, structured, and easy to follow. However, there are a few shortcomings with respect to references, and including them will improve the manuscript.

Many of the statements here are backed up with appropriate references, but some are missing. I discuss them below:

Line 74-83, Line 90-95: ‘Live attenuated vaccines have been successfully produced for decades to prevent the most dangerous diseases such as tuberculosis, measles, mumps, rubella, chickenpox or rotavirus infection. A live attenuated vaccine usually produces immunity against all proteins of the target pathogen and therefore might be more effective against various genetic variants of the pathogen. However, live vaccines carry a certain risk of causing active disease in particularly vulnerable people, i.e. those with compromised immune systems.’

Appropriate references have been added.

Line 435-439: ‘COVID-19 mRNA vaccines are the first vaccines based on 435 the mRNA technology that have been approved for use in public health programs, although studies of this technology have been conducted for several years and a few potential vaccines against other infectious diseases (e.g. CMV, Zika, influenza, rabies, malaria) have reached the second phase of testing.’

Appropriate references have been added.

Line 458: [33], [Sanofi, 2020].?

The appropriate reference has been added.

The perspectives and conclusions section need substantial revision focusing on the findings of this systematic review only. The authors need to expand upon the perspectives part more.

This part has been expanded.

Reviewer 2 Report

The authors have made an interesting attempt at “COVID-19 vaccines over three years after the outbreak of the COVID-19 epidemic.” The manuscript is interesting; however, the authors need to justify the scientific writing manuscript. Some of the general comments are provided below:

1.     Based on the review, how does the Covilo vaccine's efficacy compare to other inactivated COVID-19 vaccines, and what factors might contribute to variations in vaccine efficacy among different regions?

2.     How do protein subunit vaccines differ from live and inactivated vaccines in terms of their mode of action and immune system stimulation?

3.     What are the key advantages of using protein subunit vaccines, such as Covilo, Nuvaxovid, and VidPrevtyn Beta, in terms of safety and risk mitigation compared to other types of COVID-19 vaccines?

4.     Could the low immunogenicity of protein subunit vaccines be a limiting factor in their efficacy, and how are adjuvants used to enhance their immune response?

5.     How does the review address the potential differences in vaccine efficacy against emerging SARS-CoV-2 variants, particularly Omicron, for protein subunit vaccines compared to other vaccine types?

6.     How does the efficacy of Comirnaty (BNT162b2) mRNA vaccine from Pfizer/BioNTech compare to other mRNA vaccines, such as Spikevax (mRNA-1273) from Moderna, and how effective are these vaccines against different variants of SARS-CoV-2, including Omicron?

7.     How do the modified versions of mRNA COVID-19 vaccines, targeting specific variants like Omicron BA.1 and BA.4/BA.5, perform in terms of efficacy and immune response compared to the original vaccines?

8.     Could the use of replication-deficient viral vectors potentially limit the expression of the vaccine antigen, leading to reduced efficacy over time?

9.     Has the review discussed the potential use of vector-based vaccines as booster doses, and what evidence supports their efficacy in enhancing immunity against COVID-19 after primary vaccination?

10. Have there been any reports of adverse events or safety concerns associated with the COVID-19 vaccines that have been approved through the accelerated development and licensing procedures?

11. Has the review addressed the potential challenges and considerations for achieving global vaccine coverage during the COVID-19 pandemic, particularly in low- and middle-income countries?

12. Are there any specific guidelines or recommendations from the World Health Organization (WHO) or other international bodies regarding the monitoring and surveillance of adverse events related to COVID-19 vaccines?

13. What are the ongoing efforts and future plans to continuously monitor the safety and efficacy of COVID-19 vaccines, especially in light of emerging variants and potential long-term effects?

Author Response

Dear Reviewer,

Thank you for review of the manuscript and the helpful and important suggestions. Below we present answers for all your suggestions.

Kind regards,

Aleksandra Zasada

Reviewer 2:

Comments and Suggestions for Authors

The authors have made an interesting attempt at “COVID-19 vaccines over three years after the outbreak of the COVID-19 epidemic.” The manuscript is interesting; however, the authors need to justify the scientific writing manuscript. Some of the general comments are provided below:

  1. Based on the review, how does the Covilo vaccine's efficacy compare to other inactivated COVID-19 vaccines, and what factors might contribute to variations in vaccine efficacy among different regions?

Description of factors that might contribute to variations in vaccine efficacy has been added in the part “5. Perspectives and conclusions” together with the table showing comparison of inactivated vaccines’ efficacy (including Covilo vaccine).

  1. How do protein subunit vaccines differ from live and inactivated vaccines in terms of their mode of action and immune system stimulation?

Unlike a live attenuated vaccine, the immune response to a subunit vaccine is mostly antibody production with little or no cellular immunity results, whereas the immune response to a live attenuated vaccine is virtually identical to that produced by a natural infection. This information has been added in the manuscript.

  1. What are the key advantages of using protein subunit vaccines, such as Covilo, Nuvaxovid, and VidPrevtyn Beta, in terms of safety and risk mitigation compared to other types of COVID-19 vaccines?

According to the classification used in the manuscript, the Covilo vaccine is an inactivated vaccine and Nuvaxovid and VidPrevtyn Beta are protein subunit vaccines. Advantages of protein subunit vaccines are briefly described at the beginning of the part “2.2 Protein subunit vaccines” of the manuscript. The information about the use of Nuvaxovid as a booster has been added.

  1. Could the low immunogenicity of protein subunit vaccines be a limiting factor in their efficacy, and how are adjuvants used to enhance their immune response?

Yes, it could. The low immunogenicity of this type of vaccines is mentioned at the beginning of the part “2.2 Protein subunit vaccines” of the manuscript as well as the necessity to use adjuvants. The adjuvants used in COVID-19 vaccines includes: Matrix-M™ consists of Fraction-A and Fraction-C of Quillaja saponaria Molina extract (Nuvaxovid), AS03 composed of squalene, DL-α-tocopherol and polysorbate 80 (VidPrevtyn Beta), aluminum hydroxide gel (Abdala), aluminum hydroxide (Soberana 02, EpiVacCorona), CpG 1018 and aluminum hydroxide (MVC-COV1901), dimeric form of the RBD and aluminum hydroxide (Zifivax), CpG1018 and aluminum hydroxide  (Corbevax), SQBA adjuvant contains squalene, polysorbate 80, sorbitan trioleate, sodium citrate and citric acid (Biomervax). The adjuvants are described in the manuscript part presenting each of the specific vaccines, respectively.

  1. How does the review address the potential differences in vaccine efficacy against emerging SARS-CoV-2 variants, particularly Omicron, for protein subunit vaccines compared to other vaccine types?

The efficacy against emerging SARS-CoV-2 variants, particularly Omicron, for protein subunit vaccines is mentioned in the description of each approved vaccine. In the available studies concerning the vaccines efficacy the subunit vaccines are usually compared to mRNA vaccines. For example, in a study comparing the immune response of VidPrevtyn Beta (protein subunit vaccine) and Comirnaty (mRNA vaccine), the booster vaccine VidPrevtyn Beta was shown to be as effective as Comirnaty also against the subvariant Omicron BA.1, a study comparing the immune response of Bimervax showed effectiveness against the ancestral Wuhan-Hu-1 strain, Beta and Delta variants and Omicron BA.1 subvariant to the ancestral Wuhan-Hu-1 strain, Beta and Delta variants and Omicron BA.1 subvariant similar to that of Comirnaty (mRNA vaccine).

  1. How does the efficacy of Comirnaty (BNT162b2) mRNA vaccine from Pfizer/BioNTech compare to other mRNA vaccines, such as Spikevax (mRNA-1273) from Moderna, and how effective are these vaccines against different variants of SARS-CoV-2, including Omicron?

The efficacy of Comirnaty (BNT162b2) and Spikevax (mRNA-1273) is mentioned in the description of the vaccines respectively. The information concerning effectiveness against different variants of SARS-CoV-2 has been added.

  1. How do the modified versions of mRNA COVID-19 vaccines, targeting specific variants like Omicron BA.1 and BA.4/BA.5, perform in terms of efficacy and immune response compared to the original vaccines?

The information has been added.

  1. Could the use of replication-deficient viral vectors potentially limit the expression of the vaccine antigen, leading to reduced efficacy over time?

Yes, it could. The description of non-replicating i replicating viral vectors has been expanded.

  1. Has the review discussed the potential use of vector-based vaccines as booster doses, and what evidence supports their efficacy in enhancing immunity against COVID-19 after primary vaccination?

Yes, it has. The information concerning the use of vector-based vaccines as booster doses has been added, specifically for Vaxzevria and Jcovden.

  1. Have there been any reports of adverse events or safety concerns associated with the COVID-19 vaccines that have been approved through the accelerated development and licensing procedures?

Yes, there have. The vaccines safety and effectiveness is continuously monitored and any new adverse events are analysed and the information is added to the product information. The information about continuous monitoring of the vaccines safety has been added in the manuscript.

  1. Has the review addressed the potential challenges and considerations for achieving global vaccine coverage during the COVID-19 pandemic, particularly in low- and middle-income countries?

 Yes, it has. This information has been added in the part “4. Vaccinated population”.

  1. Are there any specific guidelines or recommendations from the World Health Organization (WHO) or other international bodies regarding the monitoring and surveillance of adverse events related to COVID-19 vaccines?

 Yes, there are. This information has been added in the part “3. Vaccines quality and safety”.

  1. What are the ongoing efforts and future plans to continuously monitor the safety and efficacy of COVID-19 vaccines, especially in light of emerging variants and potential long-term effects?

Emerging of new SARS-CoV-2 variants is monitored as well as their impact on vaccines efficacy, according to the recommendation of international public health agencies, such as WHO, ECDC, FDA. Special guidelines for variant strains update to COVID-19 vaccines have also been developed. This information has been added in the part “3. Vaccines quality and safety”.

Round 2

Reviewer 2 Report

The authors have addressed my queries and now the manuscript is acceptable for publication.